# A framework for clinical cancer subtyping from nucleosome profiling of cell-free DNA

Anna-Lisa Doebley [1,2,3], Minjeong Ko[1], Hanna Liao [2,4], A. Eden Cruikshank [1,2], Katheryn Santos[5], Caroline Kikawa[3], Joseph B. Hiatt[1,6], Robert D. Patton [1], Navonil De Sarkar[1], Katharine A. Collier [7], Anna C. H. Hoge [1], Katharine Chen [2], Anat Zimmer[1], Zachary T. Weber [7], Mohamed Adil[1,8], Jonathan B. Reichel [1,8,9], Paz Polak [10], Viktor A. Adalsteinsson [11], Peter S. Nelson [1,4,6,8,9], David MacPherson [1,4], Heather A. Parsons [5], Daniel G. Stover [7] & Gavin Ha [1,4,9] ✉

Cell-free DNA (cfDNA) has the potential to inform tumor subtype classification and help guide clinical precision oncology. Here we develop Griffin, a framework for profiling nucleosome protection and accessibility from cfDNA to study the phenotype of tumors using as low as 0.1x coverage whole genome sequencing data. Griffin employs a GC correction procedure tailored to variable cfDNA fragment sizes, which generates a better representation of chromatin accessibility and improves the accuracy of cancer detection and tumor subtype classification. We demonstrate estrogen receptor subtyping from cfDNA in metastatic breast cancer. We predict estrogen receptor subtype in 139 patients with at least 5% detectable circulating tumor DNA with an area under the receive operator characteristic curve (AUC) of 0.89 and validate performance in independent cohorts (AUC = 0.96). In summary, Griffin is a framework for accurate tumor subtyping and can be generalizable to other cancer types for precision oncology applications.

Accurate cancer diagnosis and subtype classification are critical for guiding clinical care and precision oncology. Current approaches to determine tumor subtype require a tissue biopsy, which is often difficult to obtain from patients with metastatic cancer. Therefore, at the time of recurrence or metastatic cancer diagnosis, treatment options may often be informed by clinical diagnostics from the primary tumor. However, molecular changes in the tumor can emerge during metastatic progression and in the context of therapeutic resistance. Moreover, surveying molecular changes is challenging because repeated biopsies are problematic and not routine in clinical practice for solid tumors.

Cell-free DNA (cfDNA) is DNA released into circulation by cells during apoptosis and necrosis[1]. In patients with cancer, a portion of this cfDNA is released from tumor cells, called circulating tumor DNA (ctDNA). The analysis of ctDNA can address the challenges in tissue accessibility and has demonstrated great potential for clinical utility[2–9]. Much of the current research and clinical efforts have focused on the detection of genetic alterations in ctDNA. Shallow coverage sequencing of cfDNA, including ultra-low pass whole genome sequencing (ULP-WGS, 0.1×), provides a cost-effective and scalable solution for estimating the tumor fraction (fraction of the cfDNA that is tumor

[1]Division of Public Health Sciences and Human Biology, Fred Hutchinson Cancer Center, Seattle, WA, USA. [2]Molecular and Cellular Biology Graduate Program, University of Washington, Seattle, WA, USA. [3]Medical Scientist Training Program, University of Washington, Seattle, WA, USA. [4]Department of Genome Sciences, University of Washington, Seattle, WA, USA. [5]Dana-Farber Cancer Institute, Boston, MA, USA. [6]Division of Medical Oncology, Department of Medicine, University of Washington, Seattle, WA, USA. [7]Ohio State University Comprehensive Cancer Center, Columbus, OH, USA. [8]Laboratory Medicine and Pathology, University of Washington, Seattle, WA, USA. [9]Brotman Baty Institute for Precision Medicine, Seattle, WA, USA. [10]Department of Oncological Sciences, Icahn School of Medicine, Mount Sinai, New York, NY, USA. [11]Broad Institute of MIT and Harvard, Cambridge, MA, USA. ✉e-mail: gha@fredhutch.org

derived) from the analysis of genomic copy number alterations[10–13]. Sequencing analysis of genomic alterations from ctDNA have helped to distinguish molecular subsets of tumors[14,15]. However, these genomic alterations, including somatic mutations, may not always fully explain treatment failure or identify therapeutic targets, exemplifying a major limitation of cancer precision medicine.

Tumor subtypes are often characterized by distinct transcriptional regulation, which can change during treatment resistance, leading to different clinical tumor phenotypes. For example, prostate and lung cancers may undergo trans-differentiation from adenocarcinoma to small-cell neuroendocrine phenotypes[16–20]. For metastatic breast cancer (MBC), treatment is guided based on clinical subtypes determined by the expression of the estrogen receptor (ER), progesterone receptor (PR), and human epidermal growth factor receptor 2 (HER2), often in the primary tumor[21]; endocrine therapies are prescribed to patients with ER-positive (ER +) or PR-positive (PR +) carcinomas while patients with HER2 positive tumors are prescribed anti-HER2 drugs. Patients with tumors absent for expression of all three receptors have triple negative breast cancer (TNBC) and receive chemotherapy[22]. However, receptor conversions during primary and metastatic disease progression have been frequently observed, including ~20% of patient tumors switching from ER + to ER-negative (ER-) subtypes[23–28]. Furthermore, similar to the presence of intra-tumor genomic heterogeneity in breast cancer, mixtures of clinical subtypes may also co-exist across or within metastatic lesions in the same patient, presenting major clinical challenges[29,30]. Therefore, accurate subtype classification and identification of transcriptional patterns underlying emergent clinical phenotype during therapy has critical implications for studying mechanisms of resistance and informing treatment decisions.

Recent studies have shown that the computational analysis of cfDNA fragmentation patterns from genome sequencing data can reveal the occupancy of nucleosomes in cells-of-origin[31–36]. When DNA is released into the peripheral blood following cell death, they are protected from degradation by nucleosomes[1]. At accessible genomic locations, such as at actively bound transcription factor binding sites (TFBSs) and open chromatin regions, nucleosomes are positioned in an organized manner that allows access for DNA binding proteins[37] (Fig. 1a). This nucleosome organization results in a loss of sequencing coverage, reflecting DNA degradation at the unprotected binding site with peaks of coverage at the surrounding protected locations.

Analysis of the protected and unprotected regions, termed nucleosome profiling, has been demonstrated for cancer detection and tumor tissue-of-origin prediction, including the analysis of shorter cfDNA fragments which tend to be enriched from tumor cells[38–43]. Tumor subtyping from cfDNA has been explored in castration-resistant prostate cancer (CRPC) and lung cancer by analyzing fragmentation patterns[44,45]. However, patients with other cancer types may also benefit from non-invasive subtype prediction. Specifically, predicting receptor-based subtypes from cfDNA could enable patients with late-stage breast cancer to receive targeted treatment without the need for invasive biopsies. Furthermore, current cfDNA nucleosome profiling approaches have not been optimized for ULP-WGS data. Studying the clinical phenotype of tumors from ctDNA remains challenging due to lack of robust computational methods but has obvious potential clinical benefits for guiding treatment decisions in patients with metastatic cancer.

In this present study, we develop a computational framework called Griffin to classify tumor subtypes from nucleosome profiling of cfDNA. Griffin overcomes current analytical challenges to profile the nucleosome accessibility and transcriptional regulation from the analysis of standard cfDNA genome sequencing, including ULP-WGS (0.1×) coverage. Griffin employs a GC correction procedure that is specific for DNA fragment sizes and therefore uniquely suited for cfDNA sequencing data. We apply Griffin to perform cancer detection with high

performance. Then, we demonstrate breast cancer ER subtyping from cfDNA, showing high classification accuracy and insights into tumor monitoring and heterogeneity, all achieved from analysis of ULP-WGS data. Overall, Griffin is a generalizable framework that can accurately profile chromatin accessibility from cfDNA for cancer subtype prediction and has the potential to direct personalized treatment to improve patient outcomes.

## Results

### Griffin framework for nucleosome profiling to predict tumor phenotype

We developed Griffin as an analysis framework with a GC correction procedure to accurately profile nucleosome occupancy from cfDNA. Griffin processes fragment coverage to distinguish accessible and inaccessible features of nucleosome protection (Fig. 1a). Griffin is designed to be applied to whole genome sequencing (WGS) data of cfDNA from patients with cancer to quantify nucleosome protection around sites of interest and is optimized to work for ULP-WGS data (Fig. 1b). Sites of interest can be selected from various chromatin-based assays, such as from assay for transposase-accessible chromatin using sequencing (ATAC-seq) and are tailored to address specific problems including cancer detection and tumor subtyping.

The analysis workflow begins with computing the genome-wide fragment-based GC bias for each sample. Then, for the region at each individual site of interest, the fragment midpoint coverage is computed and reweighted to remove GC biases (Methods). Midpoint coverage rather than full fragment coverage is used because it produces higher amplitude nucleosome protection signals (Supplementary Fig. 1a). Next, a composite coverage profile is computed as the mean of the GC-corrected coverage across the set of sites differential for a tissue type, tumor type, transcription factor (TF), or any phenotypic comparison of interest. By examining these coverage profiles around known cancer-specific and blood-specific TFs, we identified three quantitative features that distinguish a site as accessible and inaccessible: (a) the coverage in the window between ±30 bp (central coverage), where lower values represent increased accessibility, (b) the coverage in a window between ±1000 bp (mean coverage), and (c) the overall nucleosome peak amplitude calculated using Fast Fourier transform (amplitude). These features can be used to quantify transcription factor activity or chromatin accessibility and be used as features for detection of cancer, tumor subtyping, or studying other phenotypes of interest.

### Griffin reduces GC biases enabling detection of differential tissue accessibility

A unique aspect of Griffin is the implementation of a fragment-based GC bias correction developed by Benjamini and Speed and previously demonstrated on genomic DNA[46]. At open chromatin regions, especially at TFBSs, GC-content is non-uniform between the binding site and flanking regions, which leads to GC-related coverage biases (Fig. 2a–c, Supplementary Fig. 1b, c, Supplementary Data 1)[47]. GC bias varies between samples and between different fragment lengths within a sample[46] (Fig. 2b), which can have a major impact on nucleosome accessibility prediction (Fig. 2c). To correct for this GC bias, for each sample and each fragment length, Griffin computes the global estimated mean fragment coverage ("expected") using a fragment length position model[46] (Methods, Fig. 2b). Then, when calculating coverage around sites of interest, each fragment is assigned a weight based on the expected coverage for its GC content. This correction eliminates unexpected increases (or decreases) in coverage at binding sites, removing technical biases to enhance the tissue-associated accessibility when analyzing WGS (9–25×, Fig. 2c) cancer patient cfDNA and ULP-WGS (0.1–0.3×, Fig. 2d).

To test the performance of nucleosome profiling following Griffin GC-bias correction, we compared the estimated TFBSs accessibility

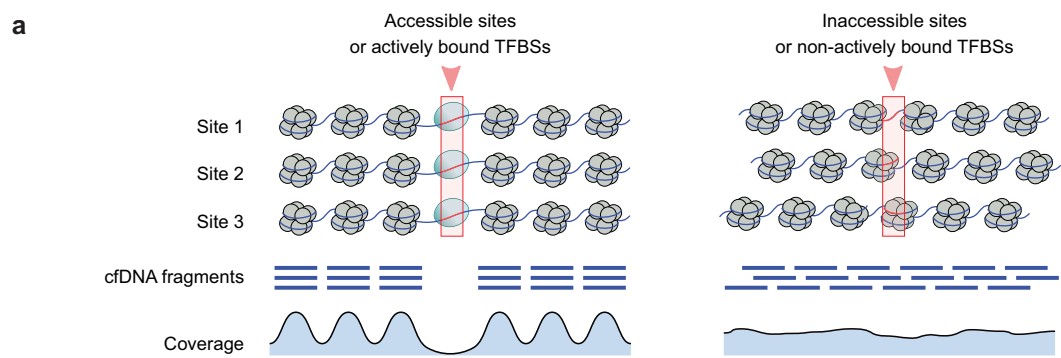

with the amount of tumor-derived DNA (i.e. tumor fraction) predicted by ichorCNA. From analysis of WGS data for 14 CRPC, two MBC, and two healthy donor samples[10,15], we observed stronger correlations between nucleosome profiles derived from shorter (35–100 bp) fragments and tumor fraction when using GC correction for multiple fragment lengths, which lead us to choose this correction strategy (Supplementary Fig. 2, Supplementary Data 2). However, in ULP-WGS

data from 191 MBC cfDNA samples[10] with ≥0.1 tumor fraction, we focused on the nucleosome sized fragments (100–200 bp) due to the low number of short fragments (<100 bp). For nucleosome sized fragments, we expected the tumor fraction to be negatively corrected with the central coverage around tumor-specific sites, and positively correlated for blood-specific sites. For a blood-specific TF, LYL1, we observed that the central coverage at TFBSs was positively correlated

**Fig. 1 | Griffin framework for cfDNA nucleosome profiling to predict cancer subtypes and tumor phenotype. a** Illustration of a group of accessible sites (left panel) and inaccessible sites (right panel), such as TFBSs. The nucleosomes (in grey) are positioned in an organized manner around the accessible sites (red box; left panel), but not around the inaccessible ones (right panel). These nucleosomes protect the DNA from degradation when it is released into peripheral blood. The protected fragments from the plasma are sequenced and aligned, leading to a coverage profile which reflects the nucleosome protection in the cells of origin. **b** Griffin workflow for cfDNA nucleosome profiling analysis. cfDNA whole genome sequencing (WGS) data with ≥0.1× coverage is aligned to hg38 genome build. (1) For each sample, fragment-based GC bias is computed for each fragment size. (2) Sites of interest are selected from any assay. Paired-end reads aligned to each site are collected, fragment midpoint coverage is counted and corrected for GC bias to produce a coverage profile. (3) Coverage profiles from all sites in a group (e.g., open chromatin for tumor subtype) are averaged to produce a composite coverage profile. Composite profiles are normalized using the surrounding region (−5 kb to +5 kb). (4) Three features are extracted from the composite coverage profile: central coverage (coverage from −30 bp to +30 bp from the site; orange 'a'), mean coverage (between −1 kb to +1 kb; green 'b'), and amplitude calculated using a Fast-Fourier Transform (FFT) (red 'c').

with tumor fraction before GC correction (Pearson's $r = 0.41$) as expected, but this correlation was much stronger after GC correction (Pearson's $r = 0.63$, Fig. 2e). For a tumor-specific TF, GRHL2, we observed a negative correlation between the central coverage and tumor fraction, as expected (Pearson's $r = −0.62$, Supplementary Fig. 3a). The mean coverage and amplitude features are also correlated to tumor fraction but appeared to be less influenced by GC bias (Supplementary Fig. 3a, b, Supplementary Data 3). Similar correlations between nucleosome profile features and tumor fraction following GC correction were also observed for blood and cancer specific DNase I hypersensitivity sites (DHSs) (Supplementary Fig. 3a).

To quantify whether GC correction reduces signal variability between samples, we examined the central coverage in the 191 MBC cfDNA ULP-WGS samples for 377 TFs in the Gene Transcription Regulation Database (GTRD)[44,48]. For each factor, we compared the variability between the central coverage and tumor fraction using the root mean squared error (RMSE) from a linear regression fit before and after GC correction. For LYL1, the RMSE decreased (0.062 to 0.046), indicating less inter-sample variation in the data after GC correction (Fig. 2e). Similarly, for 351 (93.1%) TFs, the RMSE was decreased after GC correction, indicating reduced inter-sample variability after accounting for the correlation between tumor fraction and central coverage (two-sided Wilcoxon signed rank test $p = 1.0 \times 10^{-58}$, test statistic = 1421, Fig. 2f, Supplementary Fig. 1d, Supplementary Data 3). Next, in the cfDNA samples, we systematically analyzed differentially expressed TFs between blood cells and breast cancer (Methods, Supplementary Data 4). We found that central coverage and tumor fraction were correlated for a subset of these TFs (11 of 35 cancer and 15 of 22 blood TFs, Pearson correlation two-sided adjusted $p$-value < 0.05 after GC correction), most correlations were in the expected direction, and that these correlations increased for blood TFs after GC correction (two-sided Wilcoxon signed rank test $p = 0.0013$, Supplementary Fig. 4a).

Additionally, we examined the central coverage for the 377 TFs in a cohort of 215 healthy donors[38] before and after GC correction. Because healthy donor samples have no tumor content, we evaluated the mean absolute deviation (MAD) for each TF to compare inter-sample variability. We found that the MAD for central coverage decreased after GC correction for 365 (96.8%) TFs (two-sided Wilcoxon signed rank test $p = 6.28 \times 10^{-62}$, test-statistic = 466, Fig. 2g, Supplementary Fig. 3c, Supplementary Data 5), indicating lower inter-sample variability for nearly all TFs. Finally, we tested the impact of mappability biases and copy number alterations (CNA) and found that explicit correction accounting for these factors did not improve RMSE values in the MBC cfDNA samples (Methods, Supplementary Fig. 4b–f, Supplementary Data 3). Altogether, these results suggest that the GC correction strategy in the Griffin framework reduces the variability in chromatin accessibility signals due to GC biases between samples and allows for improved detection of differential tissue accessibility in ULP-WGS data.

### Griffin analysis at TFBSs enables cancer detection

To determine if Griffin can perform cancer detection, we analyzed a published WGS (1–2×) dataset of cfDNA samples from healthy donors ($n = 215$) and early-stage cancer patients ($n = 208$) (DELFI cohort)[38]. We generated nucleosome profiles around the top TFBSs for each TF and extracted three features from each (central coverage, mean coverage, and amplitude). Due to the large number of features, we used principal components analysis (PCA) to select the top components that explained 80% of the variance (Methods). Using logistic regression on these components, we determined that the best performance was achieved when using the top 30,000 TFBSs for each of 270 TFs that contained at least this many sites (Methods, Supplementary Fig. 5a). We achieved a high performance for predicting the presence of cancer with an area under the receiver operating curve (AUC) of 0.94 (Fig. 3a, Supplementary Data 6) We observed the highest performance for stage IV cancers (AUC = 0.99) and moderately lower performance in stage I cancers (AUC = 0.93, Fig. 3a, Supplementary Fig. 5b). The performance was likely reflective of the higher tumor fractions observed in late-stage cancer relative to early-stage cancer. As anticipated, we observed higher performance for samples with tumor fraction ≥ 0.05 (AUC = 0.99) than samples with <0.03 tumor fraction (AUC = 0.92, Supplementary Fig. 6a). By cancer type, we achieved the highest performance for lung and ovarian cancers (AUC ≥ 0.99) and the lowest for pancreatic cancer (AUC = 0.85, Supplementary Fig. 6d). To test the ability to detect cancer at ULP-WGS coverage (0.1×), we applied Griffin to the same cfDNA data downsampled to 0.1× coverage and achieved an AUC of 0.89 (Fig. 3a, Supplementary Fig. 6a, d).

Next, we systematically evaluated various configurations and comparisons of Griffin for cancer detection (Supplementary Fig. 7a). First, because fragments <150 bp are enriched for tumor derived DNA[38], we tested whether different fragment size ranges, such as short (35–150 bp) or all (35–500 bp) fragments may improve our ability to detect cancer in this framework but observed a decreased performance (0.91 and 0.92 AUC, respectively, Supplementary Fig. 7a). Next, when omitting GC correction, we also observed decreased overall performance for 1–2× WGS (AUC = 0.83, Fig. 3b, Supplementary Fig. 7a) and ULP-WGS (AUC = 0.85) for all disease stages (Fig. 3b, Supplementary Fig. 7a). Then, we tested the use of mappability and copy number correction, exclusion of Griffin features, and analysis at DHSs in place of TFBSs and observed similar or lower performance (Supplementary Fig. 7a). Finally, we compared our results with the method by Ulz et al.[44], which analyzed cfDNA fragments of all lengths at TFBSs, and found it had lower performance for 1–2× WGS (AUC = 0.82) and ULP-WGS (AUC = 0.55) coverages. (Supplementary Fig. 7a, b).

To validate Griffin for the application of cancer detection, we analyzed a published cfDNA WGS (1–2×) dataset consisting of 129 lung cancer patients and 158 healthy individuals (LUCAS cohort)[45]. A validation cohort of 46 cancer patients and 385 healthy individuals was also available in this same study. There was a notable batch effect between the DELFI and LUCAS cohorts in the initial fragment size distributions and Griffin coverage profiles before and after GC correction, which prevented use of the same model on both cohorts (Methods, Supplementary Fig. 8, Supplementary Data 7). Using the 270 TFs in the Griffin analysis, we built a new model and observed an AUC of 0.76 in 1–2× WGS and 0.65 for ULP-WGS (downsampled to 0.1×)

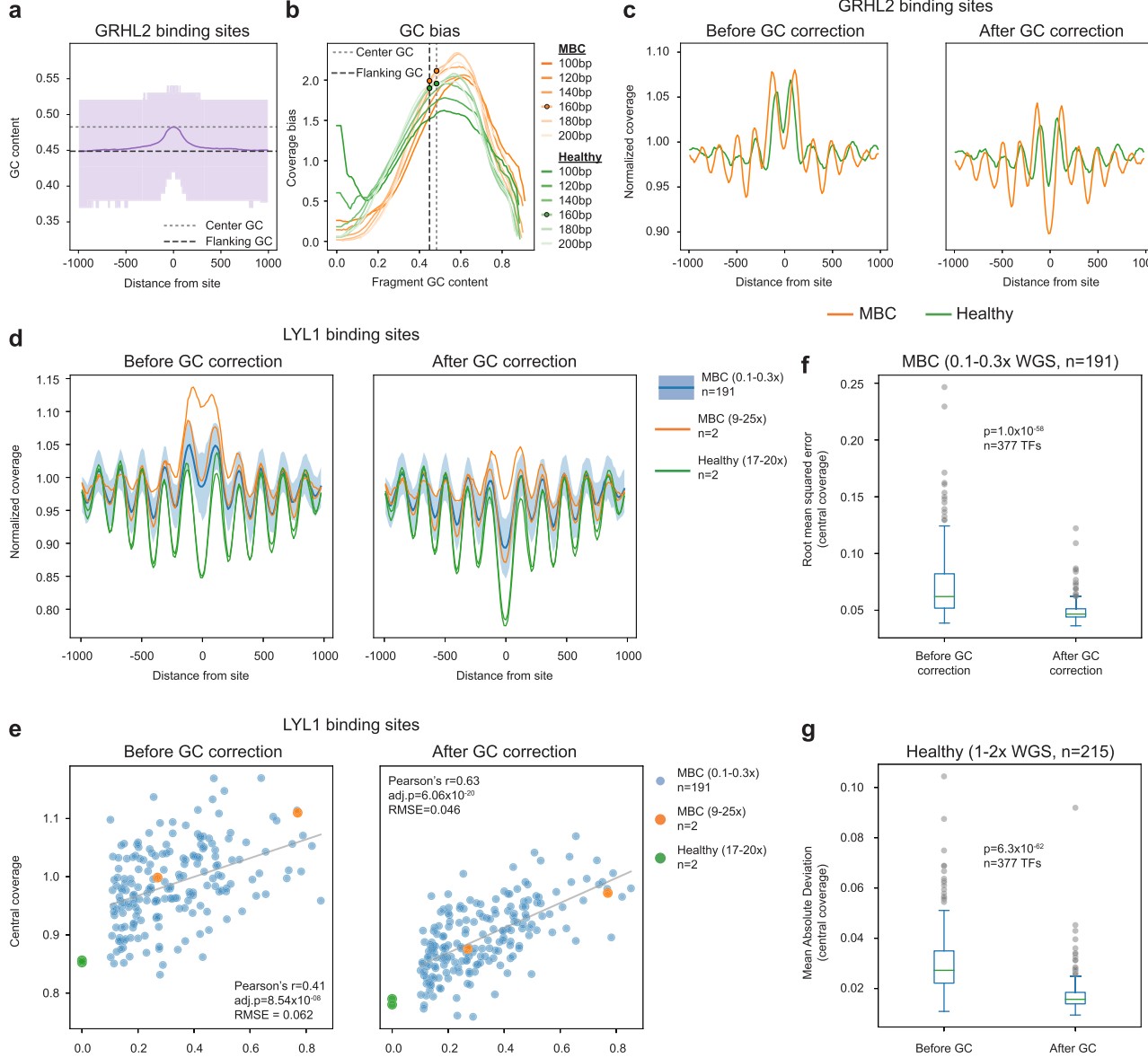

**Fig. 2 | Griffin GC bias correction improves detection of tissue specific accessibility from cfDNA. a** Mean ± IQR of GC content around 10,000 GRHL2 sites. **b** GC bias of various fragment sizes for cfDNA from a healthy donor (HD_46; green) and a metastatic breast cancer (MBC_315; orange) sample. GRHL2 center and flanking GC content are noted with dashed lines (same as [a]). The MBC sample (orange dots) has a larger difference between center (2.11) and flanking (1.99) for 165 bp fragments than the healthy sample (1.90 center, 1.96 flanking; green dots). This means that, for GRHL2, GC bias will cause increased central coverage relative to the flanking coverage and this effect will be more pronounced in the MBC sample. **c** Composite coverage profile of 10,000 GRHL2 sites before and after GC correction, shown for HD_46 and MBC_315. Before GC correction, the center has increased coverage due to GC bias. After GC correction, the MBC sample has lower central coverage, which is consistent with increased GRHL2 activity in tumor cells. **d** Composite coverage profiles of 10,000 LYL1 sites before and after GC correction, shown for two MBC

samples with deep WGS (9–25×, orange), two healthy samples (17–20×, green), and 191 MBC samples with ULP-WGS (0.1–0.3×, median ± IQR, blue). Lower central coverage in the healthy samples is consistent with LYL1 activity in hematopoiesis. **e** cfDNA tumor fraction and central coverage correlation for LYL1. GC correction increases the strength of the Pearson correlation ($n = 191$ MBC ULP-WGS samples; 2 sided with Benjamini-Hochberg FDR correction). Root mean squared error (RMSE) of the linear fit is shown. **f** Distribution of the RMSE (linear fit between central coverage and tumor fraction ($n = 191$ MBC ULP-WGS samples) across 377 TFs, before and after GC correction. Boxed range: median ± IQR, whiskers: non-outlier data (maximum extent is 1.5× IQR), grey dots: outliers. *p*-value from the Wilcoxon signed-rank test (two-sided). **g** Distribution of the mean absolute deviation (of the central coverage across 215 healthy donors [1–2× WGS]) for 377 TFs, before and after GC correction. Box elements are the same as **f**. *p*-value from the Wilcoxon signed-rank test (two-sided). Source data are provided as a Source Data file.

coverages in the LUCAS cohort (Fig. 3c, Supplementary Fig. 5c, Supplementary Data 8). We observed an AUC of 0.91 for samples with ≥ 0.05 tumor fraction, which was higher than samples with 0.03–0.05 (AUC = 0.78) and <0.03 (AUC = 0.65) tumor fractions (Supplementary Fig. 6b). Applying the trained model from the LUCAS cohort to the LUCAS validation cohort, we achieved an AUC of 0.86 across all stages, including an AUC of 0.83 for stage I cancers (Fig. 3d, Supplementary

Fig. 5d, Supplementary Data 9). The performance was 0.87 and 0.81 AUC for tumor fractions of <0.03 and ≥ 0.03, respectively (Supplementary Fig. 6c). For ULP-WGS coverage, the performance was 0.69 AUC for stage I and 0.69 AUC across all stages (Fig. 3d, Supplementary Fig. 5d). Overall, while cancer detection has been demonstrated from nucleosome profiling analysis in ctDNA[38,43–45], we show that Griffin may also be applied in this setting.

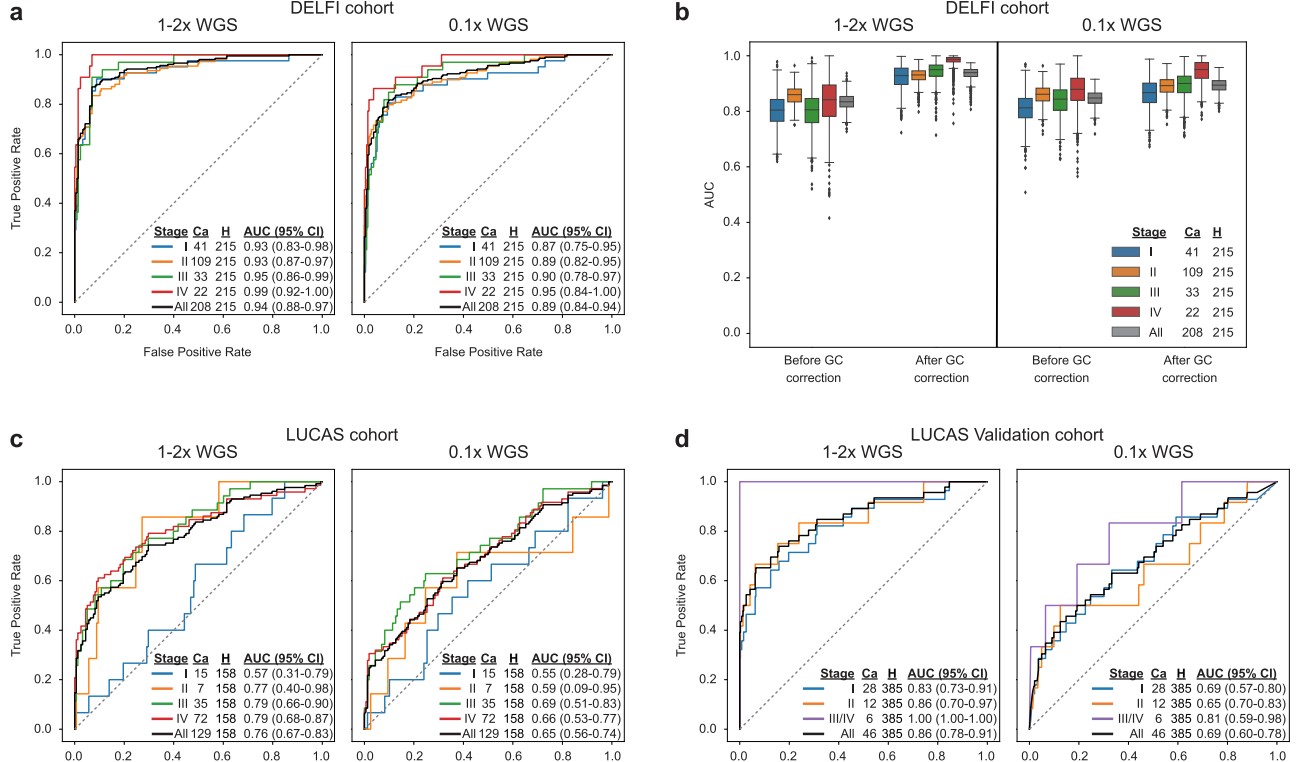

**Fig. 3 | Griffin enables accurate cancer detection.** Receiver operator characteristic (ROC) curves for logistic regression classification of cancer vs. healthy controls in three cohorts. Logistic regression was performed on the top PCA components which explained 80% of the variance in the features (central coverage, mean coverage, and amplitude) extracted from nucleosome profiles around 30,000 TFBSs for each of 270 TFs. ROC and area under the ROC curve (AUC) performance is shown for each disease stage. The number of cancer samples (Ca) is indicated for each stage. Each ROC curve also includes all healthy controls (H) from that cohort. 95% confidence intervals (CI) were obtained from 1000 bootstrap iterations. **a** Performance for DELFI cohort[38] consisting of plasma samples for 208 early-stage cancers and 215 healthy controls. **b** Comparison of the performance in the DELFI cohort before and after GC correction using Griffin. Samples are the same as in **a**. Boxplots indicate median, interquartile range (IQR), whiskers for 1.5× IQR, and outliers. **c** Performance of the LUCAS cohort[45] consisting of plasma from 129 lung cancer patients and 158 healthy patients. **d** Performance of the LUCAS validation cohort[45] consisting of plasma for 46 lung cancers and 385 healthy controls. For each dataset, performance is shown for both the original low pass (1–2×) WGS and ultra-low pass (0.1×) WGS generated by in-silico downsampling. Source data are provided as a Source Data file.

## Griffin enables accurate prediction of breast cancer subtypes from ultra-low pass WGS

Breast cancer tumor classification relies on accurate clinical determination of hormone receptor status primarily by immunohistochemistry (IHC) to quantify the expression of ER, but no ctDNA approach exists for this application. We set out to determine whether Griffin can be used to predict ER subtype status from ULP-WGS (0.1x) of cfDNA from MBC patients. We analyzed 254 samples with tumor fraction greater than 0.05 from 139 patients[10,11]. First, we inspected the Griffin profiles at TFBSs for key factors, including ESR1, FOXA1, and GATA3, which are known to be associated with ER positive tumors[49]. We observed that these TFBSs were more accessible in cfDNA samples from patients with ER + metastases compared to ER-; central coverage was significantly lower in ER + samples after accounting for tumor fraction (ANCOVA FDR adjusted $p$-value < $3.8 \times 10^{-2}$, Supplementary Fig. 9, Supplementary Data 10). To predict ER status, we initially built a logistic regression classifier using features from the Griffin profiles for all 270 TFs and achieved an accuracy of 0.71 (AUC of 0.79, Supplementary Fig. 10). We also used TFBSs features computed by the Ulz method for ER subtyping and observed an accuracy of 0.53 (AUC = 0.55, Supplementary Fig. 10), likely because it was not designed for ULP-WGS data.

Next, we used a more tailored site selection approach by analyzing regions of differential chromatin accessibility. Using ATAC-seq data generated from 44 ER + and 15 ER- primary breast tumors by The

Cancer Genome Atlas (TCGA)[50], we identified open chromatin sites that were differentially accessible between ER subtype (Methods, Fig. 4a, Supplementary Fig. 11, Supplementary Data 11–12). ER + sites ($n = 28,170$) were enriched for the TFBSs of ESR1, PGR, FOXA1 and GATA3, and ER- sites ($n = 41,712$) were enriched for the TFBSs of STAT3 and NFKB1 (Supplementary Data 13). We observed differences in coverage profiles between differential sites that were shared (9930 ER +, 22,365 ER−) and not shared (18,240 ER +, 19,347 ER−) with accessible chromatin in hematopoietic cells[51] and analyzed them separately (Fig. 4b, Supplementary Fig. 12). We applied Griffin to profile nucleosome accessibility at these four sets of ER differential accessible chromatin sites, extracting a total of 12 features. We built a logistic regression classifier to predict ER subtype from these chromatin accessibility features (Fig. 4c, Supplementary Data 14, Methods). We achieved an overall accuracy of 0.81 (AUC = 0.89, $n = 139$) with a higher performance for samples having high tumor fraction (accuracy 0.86, AUC = 0.92, $n = 101$, tumor fraction ≥ 0.1) compared to those with lower tumor fraction (accuracy 0.69, AUC = 0.75, $n = 38$, tumor fraction 0.05 to 0.1) (Fig. 4d). Systematic evaluation of different configurations and comparisons with Griffin, including fragment size ranges and data correction strategies, resulted in similar or lower performance (Supplementary Fig. 10, Methods).

We validated the trained model from the MBC dataset by evaluating its performance on independent cohorts consisting of additional ULP-WGS data or data obtained from published studies[52,53] (Methods).

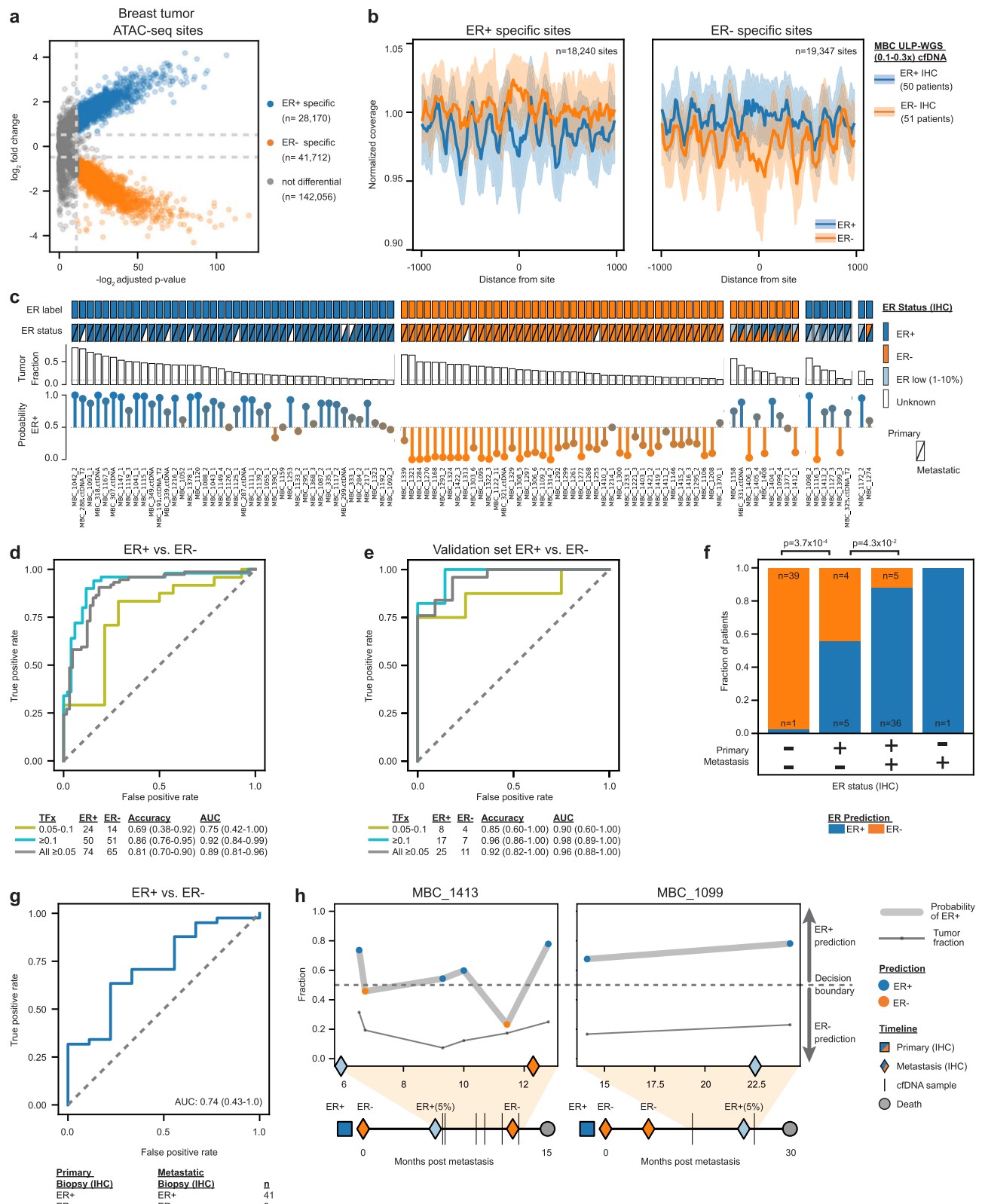

Using PCA, we did not observe batch effects between the cohorts, but rather signals could be attributed to the known ER status (by metastatic tumor IHC) and estimated tumor fraction (Supplementary Fig. 13a). In 36 patients (25 ER + , 11 ER−) with tumor fraction ≥ 0.05, we observed an overall accuracy of 0.92 (AUC = 0.96), including 0.96 accuracy (AUC = 0.98) for samples with higher tumor fraction (≥0.1, $n = 24$) and 0.85 accuracy (AUC = 0.90) for lower tumor fraction

(0.05−0.1, $n = 12$) (Fig. 4e, Supplementary Fig. 13b, c, Supplementary Data 15). For samples with tumor fraction <0.05 ($n = 35$), the accuracy was 0.54 (AUC = 0.39), indicating the lower limit of accurate ER classification is likely 0.05 tumor fraction (Supplementary Fig. 13b). These results illustrate the utility of using chromatin accessibility for cancer subtyping from ULP-WGS data and showcase ER status prediction in breast cancer from cfDNA.

**Fig. 4 | Griffin enables accurate prediction of breast cancer estrogen receptor subtypes from ultra-low pass WGS. a** ER + and ER- open chromatin sites from assay for transposase-accessible chromatin using sequencing (ATAC-seq) in ER + ( n = 44) and ER- (n = 15) breast tumors from The Cancer Genome Atlas (TCGA)[50]. Differential sites were identified using DESeq2[82] which employs a Wald test with Benjamini-Hochberg FDR correction. Sites with an adjusted p-value $<5 \times 10^{-4}$ and a $\log_2$ fold-change >0.5 or < −0.5 (dashed lines) were considered differential and are shown in blue (ER + ) or orange (ER-). **b** Composite coverage profiles (median ± IQR) for ER + (n = 18,240) and ER- (n = 19,347) sites in MBC patients (≥0.1 tumor fraction; ER + , n = 50; ER-, n = 51). Differential sites shared with hematopoietic cells have been excluded and are shown in Supplementary Fig. 12a[51]. **c** Tumor and cfDNA characteristics for 101 MBC patients with ≥0.10 tumor fraction plotted with CoMut[87]. Statuses are from immunohistochemistry on tumor tissue. Top row: Binary ER status used for training and testing the model. Second row: primary (upper left triangle) and metastatic (lower right triangle) ER status. Third row: tumor fraction from ichorCNA[10]. Fourth row: median probability ER + predicted

across 1000 bootstrap iterations. **d** Receiver operator characteristic (ROC) curve for predicting ER status. 95% CIs from 1000 bootstrap iterations. **e** Performance of the trained model on samples from three validation cohorts. **f** Predictions in patients grouped by primary and metastatic ER status. P-values from Fisher's exact test (two-sided). **g** ROC curve for predicting ER loss among patients with a primary ER positive tumor. **h** Timelines for two patients with multiple biopsies and cfDNA samples. Top: predicted probability of ER + and tumor fraction for cfDNA samples with ≥0.05 tumor fraction and ≥0.1× coverage. Bottom: timeline in months from metastatic diagnosis. The square indicates primary ER status (timeline from primary to metastatic diagnosis is not to scale). Diamonds indicate each metastatic ER status. Patient MBC_1413 had 3 metastatic biopsies, ER- at zero months (pleural fluid), weak ER + (5%) at 5.9 months (liver), and ER- at 12.3 months (pleural fluid). Patient MBC_1099 had 3 metastatic biopsies, ER- at 0 months (bone), ER- at 7 months (liver), and ER low (5%) at 22.5 months (liver). Source data are provided as a Source Data file.

## Analysis of ER status from longitudinal cfDNA suggests potential subtype heterogeneity

To further investigate the ER predictions, we inspected the classification results for 91 patients with known primary ER status and cfDNA tumor fraction of ≥0.1 (Fig. 4c, f, Supplementary Data 14). In 40 patients who had ER− status for both primary and metastatic tumors determined by IHC, we predicted 39 (95.1%) to have ER− subtype from plasma (Fig. 4f). In 41 patients who had ER + primary and metastatic tumors, we classified 36 (85.4%) to have ER + subtype. Intriguingly, in the nine patients who had clinical primary ER + and metastatic ER− status (i.e., ER loss), five (55.6%) were predicted to be ER + , and this higher prevalence of ER + prediction was statistically significant when compared to patients with no subtype switches (ER− group, $p = 3.7 \times 10^{-4}$ and ER + group, p = 0.043, two-sided Fisher's exact test, Fig. 4f). We observed a performance of 0.74 AUC for classifying ER status among patients who had ER + primary tumor status, suggesting Griffin may have some potential to classify patients with ER loss (Fig. 4g). These results demonstrate that Griffin has relatively high performance for ER classification in MBC patients with no subtype switches but ER status prediction is more challenging for patients with subtype switches perhaps due to ER subtype heterogeneity.

To further investigate the ER status predictions and subtype heterogeneity, we examined eight patients who had ULP-WGS of cfDNA from plasma collected at different timepoints and ER expression by IHC available for one or more metastatic biopsies (Fig. 4h, Supplementary Fig. 14, Supplementary Data 16)[11,54]. As an interesting example, MBC_1413 was initially diagnosed with an ER− metastatic pleural effusion but a second biopsy of the liver metastasis revealed ER expression in 5% of cells. The initial cfDNA sample was collected 178 days after and was predicted to have ER + status (0.74 probability), in agreement with the metastatic liver biopsy. A third biopsy from the pleural fluid was ER−, which was consistent with the ER− prediction (0.23 probability) from a cfDNA sample taken 26 days prior. In another example, MBC_1009 had two ER− biopsies of the bone and liver, but a third biopsy had 5% ER expression, which was consistent with ER + predictions (>0.68 probability) for cfDNA samples taken 251 days before and 52 days after. These results suggest that Griffin may be detecting ER status changes or heterogeneity of tumor biopsies and that that subtype monitoring during therapy may be a potential application.

## Discussion

In this study, we described the development of Griffin, a framework and analysis tool for studying transcriptional regulation and tumor phenotypes. Griffin applies a fragment length specific GC-correction procedure to remove the GC biases that obscure chromatin accessibility signals in cfDNA. We demonstrated that Griffin can be used to detect cancer from low pass WGS with high accuracy. We also

developed an approach to perform ER subtyping in breast cancer from ULP-WGS of ctDNA.

Griffin is versatile and can be used for various applications in cancer. We highlighted cancer detection and tumor subtype use-cases. However, Griffin can also be used for any biological comparison where transcriptional regulation and chromatin accessibility differences can be delineated. The applications described here use TFBSs from chromatin immunoprecipitation sequencing (ChIP-seq) and accessible chromatin sites from ATAC-seq. However, Griffin differs from existing frameworks due to its ability to analyze custom sites of interest that are specific to any biological context. These sites may be obtained from external sources and different assays, such as ChIP-seq, DHS, ATAC-seq or cleavage under targets and release using nuclease (CUT&RUN). As additional epigenetic data are collected by the cancer research community, including from single-cell experiments[55,56], Griffin will be integral for advancing tumor phenotype studies from liquid biopsies.

Griffin is designed for the analysis of ULP-WGS (0.1×) of cfDNA, while other nucleosome profiling methods have focused on deeper coverage sequencing. Griffin takes advantage of analyzing the breadth of sites as opposed to individual loci, which was inspired by a similar strategy used by Ulz et al.[44]. We showed that Griffin had better performance for both detecting cancer and predicting ER status from ULP-WGS data when compared to the Ulz method, likely because of its GC bias correction strategy and versatility to analyze any set of genomic regions. We observed improved performance after GC-correction consistently for all analyses, suggesting the benefit of the approach, although this improvement was minor for ER status prediction in ULP-WGS data. While the GC correction strategy was able to reduce inter-sample variability, we found that it was not able to eliminate batch effects between datasets potentially caused by different cfDNA processing and sequencing workflows, thus preventing cancer detection models from being compatible across all datasets. However, Griffin provides a framework to extract cfDNA features, enabling users to train models on new datasets, as we showed with the LUCAS and validation cohorts. Griffin can be applied to future large prospective studies using standardized plasma collection and workflows to carefully assess the performance of cancer detection in real clinical scenarios.

Although this study focused on the analysis of ULP-WGS (0.1×) of cfDNA, Griffin is not limited to low coverage data. Increased cfDNA sequencing coverage can allow for analysis of specific gene promoters and cis-regulatory elements and may enable gene expression prediction[31]. While recent studies show the promise of cfDNA methylation and cfRNA analysis for tumor phenotype analysis and cancer detection[57–63], these analytes may be challenging to isolate from clinical specimens or require specialized assays. Overall, Griffin provides a cost-effective and scalable framework requiring only standard low

coverage WGS of cfDNA, which can be more rapidly incorporated into existing platforms to predict clinical cancer phenotypes.

A limitation of the binary ER classification (ER + or ER−) is the decreased accuracy for samples with lower tumor fraction (<10%) and a 5% limit for accurate prediction, suggesting that it may be challenging to use Griffin for early-stage and minimal residual disease settings. However, in MBC, previous reports have suggested that up to 34% of MBC patients may have at least 10% tumor fraction in plasma[10], which highlights potential utility for this disease stage. TNBC patients with cfDNA tumor fraction ≥ 10% have poorer prognosis[11] and would benefit more from tumor monitoring. It may be possible to improve performance of ER subtyping for lower tumor fraction samples with additional sequencing depth, using TFBSs identified directly from ER + /− tumors, or joint analysis of multiple cfDNA timepoints from the same patient.

The application of Griffin to predict ER status from cfDNA of MBC patients led to interesting results for patients with ER loss, suggesting potential tumor heterogeneity. Intriguingly, we noticed that for the patients with ER− tumors by IHC, ER + predictions were significantly enriched when the primary tumor was ER + . Moreover, in some patients with multiple cfDNA biopsies we observed changes in predicted ER status that might be explained by the presence of metastatic tumors with both subtypes. This subtype heterogeneity and switching would typically not be captured from a single metastatic biopsy, but our results demonstrate the possibility of using the predicted ER probability to monitor subtype status over time during therapy using ctDNA. Future studies using synchronous tumor biopsy and plasma sequencing data for more patients will be needed to establish clinical utility.

We focus our breast cancer subtyping on ER prediction because its status has important utility in predicting likely benefit to endocrine therapy[64]. While PR expression is also determined in the clinic and ER−/PR + tumors are considered hormone receptor positive, these are rare, not reproducible or less useful for prognosis[65]. In our cohort, only 2 out of 139 (1.4%) patients were ER−/PR + . HER2 overexpression is important for prognosis and determining treatment such as with trastuzumab[66]. However, we were unable to identify sufficient number of open chromatin sites that were differentially accessible between HER2 positive and HER2 negative tumors. Since ERBB2 (encodes the HER2 protein) is amplified in ~20% breast cancers, one can instead assess ERBB2 copy number amplification from ctDNA genomic analysis[53,67]. Alternatively, a model to predict PAM50 status could be useful as this may be a better indicator of prognosis than ER/PR/HER2 IHC alone[68].

In summary, the Griffin framework enables prediction of tumor phenotypes from ULP-WGS. In this study, we demonstrate the use of this framework to detect cancer in early-stage cancer patients and to predict ER status in metastatic breast cancer patients. Combined with methods for predicting tumor fraction and copy number alterations[10] Griffin joins a suite of tools for in depth analysis of ULP-WGS of cfDNA enabling cost effective, non-invasive monitoring of tumors. Griffin has the potential to reveal clinically relevant tumor phenotypes, which will support the study of therapeutic resistance, inform treatment decisions, and accelerate applications in cancer precision medicine.

## Methods

The research described in this study complies with all relevant ethical regulations. New patient data (Independent MBC Cohort) was obtained under protocols which were approved by the institutional review board of the Dana Farber Cancer Institute (DFCI-09204) or Ohio State University (2007C0066, 2018C0211). Use of additional clinical data for the previously published MBC ULP-WGS cohort was approved by an institutional review board (Dana-Farber Cancer Institute IRB protocol identifiers 05-246, 09-204, 12-431 [NCT01738438; Closure effective date 6/30/2014]). Patients in all studies provided written

informed consent for the study in which they were enrolled. See descriptions of human subjects and datasets below.

### Griffin: GC-content bias correction procedure

GC content influences the efficiency of amplification and sequencing, leading to different expected coverages (coverage bias) for fragments with different GC contents and fragment lengths. This is called GC bias and is unique to each sample. We calculated the GC bias of each bam file using an implementation of the method developed by Benjamini and Speed 2012[46] which was previously implemented in deepTools[69]. However, unlike the deepTools implementation, which assumes that all fragments have the same length, we used the 'fragment length model' which calculates a separate GC bias curve for each fragment length. This is helpful for cfDNA where different samples may have different fragment size distributions and different fragment lengths have biological significance[32].

**Mappability filtering.** Prior to performing GC bias calculation, we identified all mappable regions of the genome (as described by Benjamini and Speed and implemented in deepTools) using the Umap multi-read mappability track for 100 bp reads downloaded from UCSC genome browser[70] (https://hgdownload.soe.ucsc.edu/gbdb/hg38/hoffmanMappability/k100.Umap.MultiTrackMappability.bw). We used pybedtools (0.8.0)[71] to find the mappable regions (defined as mappability score = 1) and further excluded regions with known mapping problems including the encode unified GRCh38 exclusion list (https://www.encodeproject.org/files/ENCFF356LFX/), centromeres, fix patches, and alternative haplotypes for hg38 downloaded from UCSC table browser (https://genome.ucsc.edu/cgi-bin/hgTables).

**Multi-fragment length GC bias model and correction.** We then examined all remaining regions of the genome and, for each fragment length, counted the observed GC content of every possible fragment overlapping those positions. The observed frequencies of each GC content for each fragment length are the 'genome GC frequencies' and are specific to the genome build. We then developed the 'griffin GC bias' pipeline to compute the GC bias in a given bam file. The pipeline takes a bam file, bedGraph file of valid (mappable, non-excluded) regions, and genome GC frequencies for those regions. For each given sample, we fetched all reads aligning to the valid regions on autosomes using pysam v0.15.4 (https://github.com/pysam-developers/pysam)[72]. We counted the number of observed reads for each length and GC content, excluding reads with low mapping quality (<20), duplicates, unpaired reads, and reads that failed quality control. These read counts are the 'GC counts' for that sample. We then divided the GC counts for a sample by the GC frequencies for the genome to obtain the GC bias for that bam file and normalized the mean GC bias for each fragment length to 1, resulting in a GC bias value for every combination of fragment size and GC content (except those combinations that are never observed in the genome). We then smoothed the GC bias curves. For each fragment size we took all GC bias values for fragments of a similar length (±10 bp). We sorted these values by the GC content of the fragment to create a vector of GC bias values for similar sized fragments. We then smoothed this vector by taking the median of k nearest neighbors (where k = 5% of the vector length or 50, whichever is greater) and repeated for each possible fragment length. We then normalized to a mean GC bias of 1 for each possible fragment length (excluding GC contents that are never observed) to generate a smoothed GC bias value for every possible fragment length and GC content observed in the genome.

### Griffin: Nucleosome profiling

We designed the Griffin nucleosome profiling pipeline to perform nucleosome profiling around sites of interest. This pipeline takes a

bam file, GC bias for that bam file, and site list, and assorted other parameters described below. For a given bam file and site list, we fetched all reads in a window (−5000 to +5000 bp) around each site using pysam (excluding those that failed quality control measures). We then filtered read pairs by fragment length and selected those in a range of fragment lengths (100–200 bp for all analyses in this study unless otherwise specified). For each read pair, we determined the GC bias for the fragment and assigned a weight of $\frac{1}{GC\ bias}$ to that fragment and identified the location of the fragment midpoint. We split the site into 15 bp bins and summed the weighted fragment midpoints in each bin to get a GC corrected midpoint coverage profile (see Fig. 1b for a schematic). Next, we excluded bins that overlapped regions with known mapping problems (described in Griffin: GC-content bias correction procedure) and bins with at least one unmappable position using pyBigWig for fetching data (0.3.17). We also identified bins with extremely high coverage (10 standard deviations above the mean) and removed these bins. We repeated this for every site on the site list and took the mean of all sites (ignoring excluded bins within those sites) to generate the mean coverage profile for that site list. We then smoothed the coverage profiles using a Savitzky-Golay filter with window length 165 bp and polynomial order of 3. Finally, to make samples with different depths comparable, we normalized the coverage profile to a mean coverage of 1 across the ±5000 bp window and retained the central region (±1000 bp) for further analysis.

## Griffin: Nucleosome profile feature quantification

To quantify coverage profiles, we extracted 3 features from each coverage profile. First, we calculated the coverage value at the site (±30 bp). Second, we calculated the 'mean coverage' value ±1000 bp from the site. And third, we calculated the amplitude of the nucleosome peaks surrounding the site by using a Fast Fourier Transform (as implemented in Numpy v1.21.2[73]) on the window ±960 bp from the site and taking the amplitude of the 10th frequency term. This window and frequency were chosen due to the observed nucleosome peak spacing at an active site (190 bp) which results in approximately 10 peaks in the window ±960 bp.

## Mappability correction

To test the impact of mappability bias on Griffin profiles, we implemented a per fragment mappability bias correction. First, for each sample, we obtained an approximate coverage distribution by sampling 1000 random positions within the genome (excluding positions which overlapped regions with known mapping problems see 'Griffin: GC-content bias correction procedure') and determined the cutoff for extreme outliers >5 standard deviations above the mean. Next, we split the genome into 5Mbp segments resulting in 587 segments spanning the genome (autosomes only). For each segment, we sampled every 100th position, skipping positions with known mapping problems. At sampled positions, we obtained the mappability value from Umap multi-read mappability track for 100 bp reads (described in Griffin: GC-content bias correction procedure) and the number of reads overlapping that position (excluding unpaired reads, reads with mapping quality <20, duplicates and reads that failed quality control). Sampled positions with read counts >5 SD above the mean were excluded. After obtaining the mappability values and read counts for all sampled positions, we calculated the mappability bias for each mappability value within that 5 Mbp bin by dividing the total number of reads observed at positions with a given mappability by the total number of positions with that mappability value. We repeated this procedure for all bins. Finally, we took the mappability biases for all mappability values in all bins and smoothed them using loess regression as implemented in python statsmodels (version 0.13.2)[74]. When calculating coverage profiles, we calculated the mappability value for each fragment as the mean mappability of all positions covered by the forward and reverse read. We then assigned a weight of $\frac{1}{Mappability\ bias}$ to

that fragment and multiplied this by the weight from GC bias $\frac{1}{GC\ bias}$ to get the total fragment weight used when calculating the mappability and GC bias corrected coverage profiles. This correction did not improve performance of any correlations or models and was not used in the final Griffin models.

## Copy number alteration (CNA) correction

To assess whether CNA correction improved Griffin performance, we performed CNA correction at each site prior to merging sites into composite coverage profiles. This correction was performed by dividing the coverage at each position in the profile by the mean coverage in the surrounding ±50 Kbp window. We found that the addition of CNA correction had a minimal impact on coverage profiles and did not improve the correlations to tumor fraction or performance of the cancer detection model and resulted in only a small improvement in the ER status prediction model. We did not use CNA correction in our final Griffin models, however we did leave an option to turn it on for future users who might find it useful.

## Single fragment length GC correction

In order to assess whether to use a single fragment length model or a multiple fragment length model was better able to correct GC biases around accessible sites in cfDNA, we implemented a GC correction model that assumes a single fragment length (165 bp) for all fragments similar to the method implemented by deepTools. This model used the same procedure as described in Griffin: GC-content bias correction procedure, with a few modifications. When calculating the GC counts, it assumed that every read had a fragment length of 165 bp (starting from the read start position). The resulting GC counts were then divided by the GC frequencies for 165 bp to generate the GC biases for each GC possible GC content for 165 bp fragments. Next, when generating coverage profiles, we found the GC content of each fragment and then found the GC bias for the 165 bp fragment with the most similar GC content and used this value to reweight the fragment. This single fragment length procedure was found to not perform as well for short (35–100 bp) fragments (Supplementary Fig. 2a–c) and perform similarly for nucleosome sized (100–200 bp) fragments (Supplementary Fig. 2d–f, Supplementary Fig. 7a, Supplementary Fig. 10). Consequently, the multi-fragment length model was used for all subsequent analysis.

## Early-stage cancer and healthy donor cfDNA samples

**DELFI cohort.** Whole genome sequencing (WGS) cfDNA from patients with various types of early stage cancer and healthy donors were obtained from an existing dataset published in Cristiano et al.[38]. Bam files were downloaded from EGA (dataset ID: EGAD00001005339). This data consisted of 1–2× low pass whole genome sequencing from 100 bp paired end Illumina sequencing reads. For our analyses, we used a subset of samples with 1–2× WGS of cfDNA from 208 cancer patients with no previous treatment and 215 healthy donors. These were the same samples used for the cancer detection analysis in the original Cristiano et al. study. cfDNA tumor fraction was estimated using ichorCNA (github commit 15B1D336)[10]. An hg38 panel of normals (PoN) with a 1 mb bin size was created using all 215 healthy donors in the dataset. ichorCNA was then run on all cancer and healthy samples to estimate tumor fraction. ichorCNA_fracReadsInChrYForMale was set to 0.001. Defaults were used for all other settings.

**LUCAS cohort and LUCAS validation cohort.** Whole genome sequencing (WGS) cfDNA from a prospective study of patients with lung cancer and without cancer were obtained from an existing dataset published by Mathios and colleagues[45]. Bam files were downloaded from EGA (dataset ID: EGAD00001007796). This data consisted of 1–2× low pass whole genome sequencing from 100 bp paired end Illumina sequencing reads. For our analyses, we used the subset of

samples described in the paper as the 'LUCAS' cohort and a second subset of samples described as the LUCAS validation cohort. The LUCAS cohort included 158 patients who had no history of cancer and no future cancer diagnosis and 129 patients who were diagnosed with lung cancer within days of blood draw (0–44 days). The LUCAS validation cohort included 46 patients with cancer and 385 patients without cancer. All samples were realigned to hg38 as described below in 'sequence data processing'. Tumor fraction was determined using ichorCNA (as described for the DELFI cohort) with a new panel of normals constructed using 54 separate healthy donor samples (not included in either the LUCAS or LUCAS validation cohorts) from the LUCAS study.

### Metastatic breast cancer (MBC) cfDNA samples

**Sequencing data.** WGS of cfDNA from patients with metastatic breast cancer (MBC) and healthy donors were obtained from an existing dataset published by Adalsteinsson and colleagues[10]. Bam files were downloaded from dbGaP (accession: phs001417.v1.p1). This data consisted of ~0.1× ultra-low pass whole genome sequencing (ULP-WGS) from 100 bp paired end Illumina sequencing reads. For our analyses, we used a subset of 254 ULP samples with >0.1× coverage WGS, > 0.05 tumor fraction and known estrogen receptor (ER) status. Of these 254 samples 133 were ER positive (from 74 unique patients) and 121 were ER negative (from 65 unique patients). Coverage and tumor fraction metrics were obtained from the supplementary data in the publication[10]. Additionally, we used two deep (9–25×) WGS from two MBC patients (MBC_315 and MBC_288) from the same source and two deep (17–20×) WGS from two healthy donors (HD45 and HD46) from the same source for designing and demonstrating the pipeline.

**Human subjects and clinical data.** Primary and metastatic ER status was determined by immunohistochemistry and obtained from pathological review. Metastatic survival time was also abstracted from the medical records. Use of this data was approved by an institutional review board (Dana-Farber Cancer Institute IRB protocol identifiers 05–246, 09–204, 12–431 [NCT01738438; Closure effective date 6/30/2014]).

For training and assessing the ER status classifier we labeled each sample as ER + or ER− using information about the ER status from medical records. If metastatic ER status was not known, the sample was labeled according to the primary tumor ER status (20 samples from 11 patients). ER low (1–10% ER + staining) samples (15 samples from 6 patients) were labeled ER + for the purpose of the binary classifier. For eight patients (MBC_1413, MBC_1405, MBC_1399, MBC_1099, MBC_1408, MBC_331, MBC_1312, and MBC_1404) we had information about multiple metastatic biopsies, some with multiple ER statuses among the biopsies. In these cases, we used the last biopsy taken prior to the initial cfDNA collection for the purpose of training and testing the binary ER status classifier. In a partially overlapping set of 8 patients, we also had information about multiple primary biopsies, two with multiple ER statues among the primary biopsies. In these cases, we used the first ER status to determine if there had been a subtype switch (see Supplementary Data 16 for details about biopsy ER statuses, locations, and timelines).

### Metastatic breast cancer (MBC) validation cohorts

Three independent validation cohorts were used to assess the performance of the ER status prediction model, two of these were from previously published studies. The first cohort was from the study by Ahuno et al.[52], which included WGS of cfDNA from 14 breast cancer (BRCA) patients in Ghana with known ER status and ULP WGS (0.1×) sequencing (dbGaP accession: phs002387.v1.p1). ER status and tumor fraction were obtained from the publication. Samples were then realigned to hg38 as described in 'Sequence data processing'. The second cohort was from the study by Bujak et al.[53], which included

WGS of cfDNA from 27 patients with ER + MBC (NCBI BioProject accession: PRJNA578569). ER status was obtained from the publication. The third cohort was the 'Independent MBC cohort' which consisted of ULP-WGS data generated as described below (Methods: Independent MBC cohort).

Tumor fraction for the Bujak et al cohort was estimated using ichorCNA. Samples were aligned to hg19 in order to use the default panel of normal provided with ichorCNA. 'ichorCNA_fracReadsInChrYForMale' was set to 0.001 and all other parameters were defaults. For Griffin analyses, samples from this cohort were aligned to hg38 as described in 'Sequence data processing' and downsampled to 0.1× WGS as described in 'Downsampling cfDNA sequencing data to 0.1× coverage' prior to Griffin analysis.

### Independent MBC cohort

**Human subjects.** Patients were enrolled on clinical data collection and biospecimen banking protocols. Eligible patients had biopsy-proven metastatic breast cancer. Hormone receptor status was performed using Clinical Laboratory Improvement Amendments (CLIA) approved assays. Estrogen receptor (ER) positivity was defined as ≥5% of cells positive by immunohistochemistry (IHC). Human epidermal growth factor receptor 2 (HER2) negativity was defined as IHC score 0 or 1+ and/or HER2:CEP17 fluorescent in situ hybridization (FISH) ratio <2.0. HER2 positivity was defined as IHC score 3 +, or IHC score 2+ with HER2:CEP17 FISH ratio ≥2.0. Triple negative breast cancer (TNBC) was defined as <5% staining for ER and progesterone receptor (PR), as well as HER2 negativity as previously defined. The protocols were approved by the institutional review board of the Dana Farber Cancer Institute (DFCI-09204) or Ohio State University (2007C0066, 2018C0211). All patients provided written informed consent for blood sample collection, genomic analyses, and collection of clinicopathologic data. A total of 103 samples from 30 patients were used for this study. This included 15 hormone receptor positive patients and 15 TNBC patients.

**Blood sample processing and plasma extraction.** Venous blood samples (10 mL) were collected in EDTA (BD, Franklin Lakes, NJ), CellSave Preservative (Cell Search, Raritan, NJ) or Cell-Free DNA BCT (Streck, Omaha, NE) tubes. EDTA tubes were processed within 4 h of collection and Streck tubes within 48 h. Whole blood was centrifuged at 1900 g for 10 min at room temperature with the brake off. Plasma was removed and transferred to Eppendorf DNA LoBind tubes, then centrifuged at 1900 g for 10 min at room temperature. Plasma was transferred to cryovials and frozen at −80 °C for storage.

Frozen aliquots of plasma were thawed at room temperature. cfDNA was extracted using the QIAsymphony DSP Circulating DNA Kit according to the manufacturer's instructions, with ~4 mL of plasma as input and with a 60 μL DNA elution.

**Library construction.** Initial DNA input is normalized to be within the range of 25–52.5 ng in 50 μL of TE buffer (10 mM Tris HCl 1 mM EDTA, pH 8.0) according to picogreen quantification. Library preparation is performed using a commercially available kit provided by KAPA Biosystems (KAPA HyperPrep Kit with Library Amplification product KK8504) and IDT's duplex UMI adapters. Unique 8-base dual index sequences embedded within the p5 and p7 primers (purchased from IDT) are added during PCR. Enzymatic clean-ups are performed using Beckman Coultier AMPure XP beads with elution volumes reduced to 30 μL to maximize library concentration.

**Post library construction quantification and normalization.** Library quantification was performed using the Invitrogen Quant-It broad range dsDNA quantification assay kit (Thermo Scientific Catalog: Q33130) with a 1:200 PicoGreen dilution. Following quantification, each library is normalized to a concentration of 35 ng/μL, using Tris-HCl, 10 mM, pH 8.0.

**Library pool creation and ultra-low pass sequencing.** In preparation for the sequencing of the ultra-low pass libraries (ULP), approximately, 4 μL of the normalized library is transferred into a new receptacle and further normalized to a concentration of 2 ng/μL using Tris-HCl, 10 mM, pH 8.0. Following normalization, up to 95 ultra-low pass WGS samples are pooled together using equivolume pooling. The pool is quantified via qPCR and normalized to the appropriate concentration to proceed to sequencing. Cluster amplification of library pools was performed according to the manufacturer's protocol (Illumina) using Exclusion Amplification cluster chemistry and HiSeq X flowcells. Flowcells were sequenced on v2 Sequencing-by-Synthesis chemistry for HiSeq X flowcells. The flowcells are then analyzed using RTA v.2.7.3 or later. Each pool of ultra-low pass whole genome libraries is run on one lane using paired 151 bp runs.

## Castration resistant prostate cancer (CRPC) samples

Deep WGS (16-61x) of cfDNA from patients with castration resistant prostate cancer (CRPC) and healthy donors were obtained from an existing dataset published by Viswanathan and colleagues and Adalsteinsson and colleagues[10,15]. Bam files were downloaded from dbGaP (accession: phs001417.v1.p1). Coverage and tumor fraction metrics were obtained from the supplementary data in the publications. These samples were used for designing and demonstrating the pipeline.

## Sequence data processing

All sequencing data used in this study was realigned to the hg38 version of the human genome (downloaded from http://hgdownload.soe.ucsc. edu/goldenPath/hg38/bigZips/hg38.fa.gz). Bam files were unmapped from their previous alignment using Picard (v2.18.29) SamToFastq. They were then realigned to the human reference genome according to GATK best practices[75] using the following procedure. Fastq files were realigned using BWA-MEM (v0.7.17)[76]. Files were then sorted with samtools (v1.9)[77], duplicates were marked with Picard, and base recalibration was performed with GATK (v4.1.0.0), using known polymorphisms downloaded from the following locations: https://console. cloud.google.com/storage/browser/genomics-public-data/resources/ broad/hg38/v0/Mills_and_1000G_gold_standard.indels.hg38.vcf.gz and https://ftp.ncbi.nih.gov/snp/organisms/human_9606_b151_GRCh38p7/ VCF/GATK/All_20180418.vcf.gz.

## Transcription factor binding site (TFBS) selection

Transcription factor binding sites (TFBSs) were downloaded from the GTRD database[48]. This database contains a compilation of ChIP seq data from various sources. For our analyses, we used the meta clusters data (version 19.10, downloaded from https://gtrd.biouml. org/downloads/19.10/chip-seq/Homo%20sapiens_meta_clusters. interval.gz). This contains meta peaks observed in one or more ChIP seq experiments. The GTRD database contains some ChIP seq experiments for targets that are not transcription factors (TFs). These were excluded by comparing against a list of TFs with known binding sites in the CIS-BP database[78] (v2.00 downloaded from http://cisbp.ccbr.utoronto.ca/bulk.php). The site position was identified as the mean of 'Start' and 'End'. For GC, mappability, and CNA correction analyses as well as TFBSs nucleosome profiling in MBC, TFs with less than 10,000 sites on autosomes were excluded resulting in 377 TFs. For each remaining TF, the top 10,000 sites were selected by choosing those with the highest 'peak.count' (number of times that peak has been observed across all experiments). For cancer detection we tried several cutoffs (1000 to 50,000 TFBSs) and selected an optimal cutoff of 30,000 sites, resulting in 270 TFs (see number of sites analysis below). For the MBC ER status classifier shown in Supplementary Fig. 10, we also used the top 30,000 sites.

## Identification of differential TFs in blood and cancer

To identify transcription factors that were differentially expressed between blood cells and breast cancer, we used the University of California Santa cruise (UCSC) Xena online differential gene expression analysis tool (http://xena.ucsc.edu/)[79] which uses the Appyter bulk RNA-seq analysis pipeline to run Limma-Voom differential gene expression analysis[80]. After launching the tool via a web browser, we selected the publicly available 'TCGA TARGET GTEX' study which includes RNA seq from TCGA tumors as well as RNA seq from GTEX healthy tissues. The version of the data was 2016-04-12. We selected the phenotypic variables 'main category' which groups samples by tissue or tumor type and 'study' which groups samples by study. We then ran a differential gene expression analysis on the 'main category' variable and selected GTEX Blood (337 samples) and TCGA_Breast_Invasive_Carcinoma (1099 samples) as the two subgroups to compare in the analysis. All other parameters were left as defaults. We used the outputs to determine which of the 377 TFs (see 'Transcription factor binding site (TFBS) selection' above) were differentially expressed between blood cells and breast cancer cells (using default cutoffs: adjusted $p$-value ≤ 0.05 and absolute value of log2 fold-change ≥1.5). This yielded 107 TFs that were upregulated in BRCA and 82 TFs that were upregulated in blood. We noted that some TFs shared a large number of binding sites with TFs that were upregulated in the opposite tissue type. For instance, MECOM (also called EVI1) has previously shown to be more accessible in blood than in BRCA[44] and we saw the same trend of increased accessibility in blood in our data. However, according to the differential RNA seq analysis, MECOM is actually upregulated in BRCA relative to blood. We suspected that this discrepancy is due to the fact that MECOM shares almost half of its top 10,000 sites (4465 sites) with blood specific LYL1 and >15% of the top 10,000 sites each with of a number of other factors that are upregulated in blood including ZBTB16 (2884 sites), TBX21 (1837 sites), STAT5A (1936 sites), and SPI1 (2075 sites). Because of this type of extensive site overlap seen in some differential TFs, we implemented a filter to exclude differential TFs which shared too many sites with the opposite tissue type. For the top 10,000 TFBSs for each TF, we examined how many of them overlapped (binding site was within ± 250 bp) with the top 10,000 TFBSs for each TF of the opposite tissue type (i.e. for each TF that was upregulated in blood, we looked at how many sites it shared with each of the 107 TFs that were upregulated in BRCA and took the mean of these 107 values). If a TF overlapped with an average of 400 or more sites for the factors expressed in the opposite tissue type, it was excluded from the list of differentially expressed TFs because it was considered to share too many sites with the opposite class, potentially confounding tissue specific accessibility. This left us with 22 TFs that were upregulated in blood and 35 factors that were upregulated in cancer.

## DNase I hypersensitivity site selection

DNase I hypersensitivity sites for a variety of tissue types were downloaded from https://zenodo.org/record/3838751/files/DHS_Index_and_ Vocabulary_hg38_WM20190703.txt.gz[81]. These sites were split by tissue type for a total of 16 site lists. The 'summit' column was used as the site position. The sites were sorted by the number of samples where that site had been observed ('numsamples') and the top 10,000 most frequently observed sites were selected for each tissue type.

## ATAC-seq site selection for ER subtyping

Assay for transposase-accessible chromatin using sequencing (ATAC-seq) site accessibility for primary breast cancer samples from The Cancer Genome Atlas (TCGA) were downloaded from the TCGA ATAC-seq hub (https://gdc.cancer.gov/about-data/publications/ATACseq-AWG)[50]. A file containing raw counts for all cancer type specific sites were downloaded ('All cancer type-specific count matrices in raw counts') and the file containing breast cancer specific sites was used

('BRCA_raw_counts.txt'). The locations of these sites and patient metadata were obtained from the supplementary tables in the paper[50]. Sites on autosomes were kept for further analysis for a total of 211,938 sites. Differentially accessible sites between ER + ($n = 44$) and ER− ($n = 15$) tumors were identified using the DESeq2 software[82]. The software was run using default settings described in the 'quick start' guide with no co-variates. A differential accessibility experiment was run using the 'DESeq' and 'results' functions followed by log fold change shrinkage using the 'lfcShrink' function. Sites with an adjusted $p$-value $< 5 \times 10^{-4}$ were selected. Additionally, selected sites were further filtered based on the log2 fold-change between ER + and ER− tumors (see 'Selection of cutoffs for ER status differential ATAC-seq sites' below). Sites with a log2 fold change $>0.5$ were classified as ER + , while sites with a log2 fold change $< -0.5$ were classified as ER−. These site lists were further split into sites shared with hematopoietic cells and those not shared with hematopoietic cells. Hematopoietic sites were obtained from a database of single cell ATAC-seq data[51] (GEO accession number: GSE129785, peak file available here: https://ftp.ncbi. nlm.nih.gov/geo/series/GSE129nnn/GSE129785/suppl/GSE129785% 5FscATAC%2DHematopoiesis%2DAll%2Epeaks%2Etxt%2Egz). Hematopoietic sites were lifted over to hg38 using the UCSC liftover command line tool and sites that changed size during liftover (0.2% of sites) were discarded. ER differential ATAC-seq sites were overlapped with hematopoietic sites (Overlapping sites were defined as site centers being within 500 bp of one another) using pybedtools intersect[71,83]. This resulted in a total of 4 differential site lists: ER positive sites that were not shared with hematopoietic cells (18,240 sites), ER positive sites that were shared with hematopoietic cells (9930 sites), ER negative sites that were not shared with hematopoietic cells (19,347 sites), and ER negative sites that were shared with hematopoietic cells (22,365 sites).

To further characterize these sites, overlapped the four site lists with the top 10,000 sites for each of 377 transcription factors (TFs) using pybedtools intersect. An overlapping pair of sites was defined as having <500 bp between site centers. Each differential ATAC-seq site list was compared against each list of TFBSs and the total number of ATAC sites overlapping one or more TFBSs on the given list was recorded (Supplementary Data 13).

## Selection of cutoffs for ER status differential ATAC-seq sites

In order to select the optimal $p$-value and fold change cutoffs for identifying ER + and ER− differential ATAC-seq sites, we tried several different cutoffs for the values output by DESeq2. First we tried 4 different log2 fold-change cutoffs (no cutoff, 0.5, 1, 2) while holding the adjusted $p$-value cutoff constant at 0.05. Second, we tried 3 additional $p$-value cutoffs while holding the log2 fold-change constant at 0.5. For each cutoff, we ran the griffin nucleosome profiling analysis on the selected ATAC-seq sites, using 100–200 bp fragments. We then extracted features and used these in a logistic regression model to predict ER status as described below ('Machine learning, bootstrapping, and performance evaluation procedure' and 'ER status classification in the MBC cohort') and calculated the mean accuracy across all bootstraps. We found that there was a relatively small difference between cutoffs (-2% accuracy) but chose the cutoff with the highest accuracy (adjusted p-value $\leq 5 \times 10^{-4}$ and absolute value of log2 fold-change $>0.5$) for our final model.

## Quantification of GC content at TFBSs

For 377 TFs (see Transcription factor binding site (TFBS) selection above), the GC content around the top 10,000 TFBSs was quantified (Shown in Fig. 2a and Supplementary Fig. 1). The sequence at each site (±1000 bp) was fetched from the genome and the GC content was calculated. Positions within sites that overlapped the exclusion lists or had zero mappability were excluded. GC content at individual sites was then smoothed using a Savitsky-Golay filter with length 165 bp and

polynomial order zero. The mean GC content at the site center was calculated as the mean of the smoothed GC content across all sites in the window ±30 bp from the site. The mean flanking GC content was calculated as the mean of the GC content in the window ±1000 bp from the site, excluding the central region (±30 bp).

## Assessment of Griffin before and after GC correction, mappability correction, and CNA correction

**Tumor fraction correlations at TFBSs.** For 191 MBC ULP samples with >0.1 tumor fraction, nucleosome profiling with and without GC correction was performed on the top 10,000 sites for each of 377 transcription factors (TFs) using nucleosome sized fragments (100–200 bp). For each TF, the relationship between central coverage and tumor fraction was modeled using scipy.stats.linregress[84] producing a Pearson correlation (r) and line of best fit (scipy version 1.7.1). Pearson p-values for each feature type were adjusted using a Benjamini-Hochberg FDR correction. Root mean squared error (RMSE) was calculated from the line of best fit. This was performed both before and after GC correction as illustrated for LYL1 in Fig. 2e. For all 377 TFs, the RMSE values before and after GC correction were compared using a Wilcoxon signed-rank test (two-sided). This same procedure was applied to test the benefit of an additional mappability correction step and an additional copy number correction step.

**Mean absolute deviation (MAD) at TFBSs.** For 215 healthy donors, nucleosome profiling with and without GC correction was performed on the top 10,000 TFBSs for each of 377 TFs. For each TF, the MAD of the central coverage values was calculated both before and after GC correction. For all 377 TFs, the MAD values before and after GC correction were compared using a Wilcoxon signed-rank test (two-sided).

## Quantification of differential accessibility of TFBSs and ATAC sites in MBC

To determine whether nucleosome profiles around TFBSs were differentially accessible between ER + and ER− samples we performed an analysis of covariance (ANCOVA) as implemented in Pingouin (v0.5.1)[85]. For this analysis, we used the 191 MBC samples with ≥0.1 tumor fraction and ≥0.1 coverage. Nucleosome profile feature (central coverage, mean coverage, or amplitude) was the dependent variable, primary tumor status was the independent variable ('between'), and tumor fraction was a covariate. We performed this analysis on all 3 features for all 377 TFs with 10,000 or more sites. We then used Benjamini-Hochberg FDR correction to perform multi-test correction for on the p-values for ER status and tumor fraction for each feature.

We performed the same ANCOVA analysis on the features for the 4 types of ER differential ATAC sites but without FDR correction as there were only a total of 12 features (central coverage, mean coverage, and amplitude for each of the 4 site types).

## Machine learning, bootstrapping, and performance evaluation procedure

To detect cancer or predict ER subtype, we used logistic regression with Ridge regularization (i.e. L2 norm) as implemented in scikit-learn (v0.23.2)[86]. All feature values were scaled to a mean of 0 and a standard deviation of 1 prior to performing bootstrapping and fitting the models. We used the following bootstrapping procedure to train and assess the performance of our models. First, we selected n samples with replacement from the full set of n samples and used this as a training set. Samples that weren't selected were used as the test set. We then used 10-fold cross-validation on the training set to select the hyperparameter 'C' (inverse of the regularization strength). To account for class imbalances in the data we used set the 'class weight' parameter to 'balanced' to adjust the sample weighs inversely proportional to the class frequencies. We trained a final model on all the training data using the selected regularization strength. Finally, we tested this

model on the test set and recorded the performance (accuracy and AUC values) and sample probabilities. Then, a new training set was selected, and the procedure was repeated for 1000 iterations. After completing the bootstrap iterations, we calculated the AUC and accuracy from each bootstrap iteration and used these to generate the mean and 95% confidence interval around each of these values and to create boxplots. To visualize the ROC curve, we used the median probability from all bootstraps where each sample was included in the test set. For further downstream analyses, including the comut plot, barplots, and timelines we used this same median probability.

### Features used for the final cancer detection classification
To detect cancer, we applied the logistic regression approach described above and built a logistic regression classifier on features extracted from the DELFI cohort cancer and healthy samples. First, we performed nucleosome profiling in these samples (selecting fragments 100–200 bp in length). For our finalized model we used 30,000 TFBSs each for 270 TFs with at least 30,000 sites in the GTRD database (see 'Selection of number of TFBSs for cancer detection' below). We extracted three features (as described above 'Griffin: nucleosome profile feature quantification') from each coverage profile for a total of 810 features. We then scaled these features to a mean of 0 and a SD of 1. Within each bootstrap iteration, we reduced the dimensionality of the feature using PCA as implemented in scikit learn[86] on the training set and selected the features that explained 80% of the variance. We then applied this same PCA transformation to the test set for that bootstrap. These PCA components were then used as the inputs for the logistic regression model which was trained on the training set, and tested on the test set.

For the LUCAS cohort, we found that there were batch effects that prevented using the same model trained on the DELFI cohort. Because of this we trained and tested a new model on the LUCAS cohort using the same bootstrapping approach and performed a final validation of this model in the LUCAS validation cohort (described below, 'Validation of the cancer detection model').

Finally, we downsampled both the DELFI and LUCAS cohorts to ~0.1× coverage (procedure described below) and performed the same cancer detection analysis in this lower coverage data.

### Validation of the cancer detection model
After training and testing a logistic regression model using the bootstrapping approach on the LUCAS cohort, we applied the final model to the previously unseen LUCAS validation cohort (385 healthy samples and 46 cancer samples). To get this final model, first, we performed PCA on the full LUCAS cohort (not including the LUCAS validation cohort) and extracted 35 features that explained 80% of the variance in that cohort. Then, we used these 35 features to build a logistic regression model using the regularization strength most frequently chosen by the 1000 bootstraps on the LUCAS cohort ('C' = 0.01). Finally, we applied the PCA transformation and logistic regression model to the LUCAS validation cohort, extracted the same 35 features, and got a probability of cancer for each sample. We obtained confidence intervals for the AUC using a bootstrap procedure in which we selected 431 samples with replacement from the original 431 samples and calculated the AUC for the selected samples. We then repeated this 1000 times to get 1000 AUC values which we used to obtain confidence intervals and boxplots.

We repeated this same procedure for the downsampled LUCAS validation cohort.

### Selection of number of TFBSs for cancer detection
In order to select the optimal number of TFBSs for cancer detection, we tried several different cutoffs for the number of sites (1000, 5000, 10000, 20000, 30000, and 50000 sites). For each cutoff we identified all TFs with at least that many sites in GTRD resulting in 566, 446, 377,

316, 270, and 202 TFs respectively for the cutoffs above. We then picked the top sites by choosing those with the highest 'peak.count'. We next used the logistic regression with bootstrapping and PCA dimensionality reduction described above ('Features used for the final cancer detection classification') to train and test models on both the original 1–2× WGS DELFI cohort samples and the downsampled 0.1× DELFI cohort samples. We found that the number of sites had a greater impact on the downsampled data (Supplementary Fig. 5a) so we selected the cutoff with the highest AUC in downsampled data which was 30,000 sites.

### Cancer detection from DNAse hypersensitivity sites
In addition to examining TFBSs, we also performed nucleosome profiling at the 16 tissue-specific DHS site lists described above. We extracted the same 3 features from each site profile for a total of 48 features and used the same bootstrapping plus PCA dimensionality reduction described above to test the performance of this model.

### Downsampling cfDNA sequencing data to 0.1× coverage
WGS data for the DELFI cohort, LUCAS cohorts (training and validation), and Bujak et al. dataset was aligned to aligned to hg38 and subsequently downsampled using Picard DownSampleSam. The probability used by DownSampleSam was calculated based on a target of 2,463,109 read pairs which resulted in approximately 0.11x coverage as calculated by Picard CollectWgsMetrics. Downsampled bam files from the DELFI dataset were realigned to hg19 for use in the Ulz pipeline for comparison. The realignment procedure was the same as above but using the hg19 genome (downloaded from https://hgdownload.soe.ucsc.edu/goldenPath/hg19/bigZips/hg19.fa.gz) and hg19 known polymorphic sites for base recalibration (downloaded from ftp://gsapubftp-anonymous@ftp.broadinstitute.org/bundle/hg37/Mills_and_1000G_gold_standard.indels.hg37.vcf.gz and ftp://ftp.ncbi.nih.gov/snp/organisms/human_9606_b151_GRCh37p13/VCF/GATK/All_20180423.vcf.gz).

### ER status classification in the MBC cohort
To predict ER status, we applied the logistic regression approach described above to features extracted from the MBC patient samples. Because some patients had multiple samples, we modified the bootstrapping procedure to select 139 patients (rather than samples) with replacement from a full set of 139 patients. For each selected patient, all samples from that patient were added to the training set (If a patient was selected multiple times, all their samples were included multiple times). This ensured that separate samples from the same patient (biological replicates) could not appear in both the training and test set. Samples from patients that weren't selected were used as the test set.

For our model, we applied nucleosome profiling using 100–200 bp fragments to the 4 ER differential ATAC seq lists and extracted 3 features per profile for a total of 12 features. For evaluating the model, we only included the first timepoint for each patient in the test set when calculating the accuracy and AUC for each bootstrap iteration. This prevented a small number of patients with many samples from having a large impact on the scores.

### ER status prediction from TFBSs
In order to assess whether ER status could be predicted from the nucleosome profiles around TFBSs, we performed nucleosome profiling for the top 30,000 sites for 270 TFs and extracted 3 features each for a total of 810 features. We then used the bootstrapping approach described above ('ER status classification in the MBC cohort') to train and test the model. Because of the high dimensionality of the data, within each bootstrap, we performed PCA on the training set and selected the top PCA components that described 80% of the variance. We then used these components as the features in our logistic

regression model. This model did not perform as well as the differential ATAC site model and was not used for further analysis.

## Validation of the ER status prediction model

After training and testing a logistic regression model using the bootstrapping approach on the MBC cohort and features from griffin profiles around differential ATAC sites, we applied the final model to the three previously unseen validation cohorts. To get this final model, we trained a logistic regression model on the full MBC dataset (254 samples) using the regularization strength most frequently chosen by the 1000 bootstraps on the MBC cohort ('C' = 0.1). We then applied this model to the three validation cohorts and got the probability of ER + for each sample. For patients with multiple samples (in the independent MBC cohort) we used the first timepoint when evaluating performance, resulting in a total of 71 samples from unique patients across all three cohorts. We obtained confidence intervals for the accuracy and AUC using a bootstrap procedure in which we selected 71 samples with replacement from the original 71 samples and calculated the AUC and accuracy for the selected samples. We repeated this procedure 1000 times to get 1000 AUC and accuracy values which we used to obtain confidence intervals and boxplots.

## Transcription factor profiling using pipeline from Ulz et al.

We downloaded the Transcription Factor Profiling pipeline published by Ulz and colleagues from Github (https://github.com/PeterUlz/TranscriptionFactorProfiling)[44] and ran it using the following procedure as described in the paper. hg19 aligned bam files were used because the pipeline was written to for this version of the genome. Scripts were modified so that they worked in python3. We trimmed the reads in each bam to 60 bp using 'trim from bam single end' with modifications to skip unaligned reads. We ran ichorCNA on the original (untrimmed) bam using the default ichorCNA settings for hg19 except the bin size, which was modified to 50,000 bp and no panel of normals. We then ran the transcription factor profiling analysis on the trimmed bam using the script run_tf_analyses_from_bam.py with options '-calccov' and '-a tf_gtrd_1000sites' and the ichorCNA corrected depth file as the '-norm-file'. This ran transcription factor profiling on 1,000 sites for each of 504 TFs. Finally, we ran the scoring pipeline. We used the high frequency amplitude ('HighFreqRange') for each of the 504 TFs in the accessibility output file (Accessibility1KSitesAdjusted.txt) as the features for a logistic regression model using the same bootstrapping with PCA dimensionality reduction as described for cancer detection and ER status prediction from TFBSs above.

## Reporting summary

Further information on research design is available in the Nature Portfolio Reporting Summary linked to this article.

## Data availability

Data generated for the independent MBC cohort are not publicly available because patients did not consent to data deposition in public data repositories but are available under restricted access from daniel.stover@osumc.edu on reasonable request, fulfilling a data transfer agreement; data will be available for up to one year for research purposes. All other datasets used in this study were published datasets from previous studies and can be obtained by authorization through a database (WGS of cfDNA) or downloaded freely (ATAC-seq, RNA-seq, ChIP-seq, DNAse-seq). WGS of cfDNA from castration resistant prostate cancer, metastatic breast cancer, and healthy donors was obtained from dbGaP (accession phs001417.v1.p1). WGS of cfDNA from breast cancer patients in the Ahuno et al. study was obtained from dbGaP (accession phs002387.v1.p1). WGS of cfDNA from ER + breast cancer patients in the Bujak et al. study was obtained from NCBI (BioProject accession number PRJNA578569). The DELFI cohort (WGS of cfDNA from early stage cancer patients and healthy donors) was obtained from EGA (dataset ID EGAD00001005339). The LUCAS and LUCAS validation cohorts (WGS of cfDNA from lung cancer patients and healthy donors) were also obtained from EGA (EGAD00001007796). ATAC seq peak counts and sample metadata were downloaded from TCGA (https://gdc.cancer.gov/about-data/publications/ATACseq-AWG) and is freely available without authorization. DNAse-seq was downloaded from zenodo (https://zenodo.org/record/3838751/files/DHS_Index_and_Vocabulary_hg38_WM20190703.txt.gz). ChIP seq was downloaded from GTRD version 19.10 (https://gtrd.biouml.org/downloads/19.10/chip-seq/Homo%20sapiens_meta_clusters.interval.gz). RNA seq to obtain differential gene lists was accessed from https://toil-xena-hub.s3.us-east-1.amazonaws.com/download/TcgaTargetGtex_RSEM_Hugo_norm_count.gz (version 2016-04-12) using the UCSC Xena online tool. Source data are provided with this paper.

## Code availability

Griffin software and the subtype classifier tool can be obtained from https://github.com/adoebley/Griffin. Code for analysis and machine learning models can be accessed at https://github.com/adoebley/Griffin_analyses.

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

## Acknowledgements

We thank the many patients and their families for their generosity in contributing to this study. We also thank Patricia Galipeau and the Ha laboratory for helpful discussions and critical reading of the manuscript. This work was supported by the National Institutes of Health (K22 CA237746, R21 CA264383, DP2 CA280624 to G.H.), the V Foundation Scholar Grant (to G.H.), the Prostate Cancer Foundation Young Investigator Award (to G.H.), the Fund for Innovation in Cancer Informatics Major Grant (to G.H.), and the Department of Defense Idea Development Award (W81XWH-21-1-0513). This research was also supported by the NIH/NCI Cancer Center Support Grant P30 CA015704, Brotman Baty Institute for Precision Medicine, NIH grants (UH2 CA239105 to D.G.S.; P50 CA097186; R01 CA2344715 to P.S.N; K08 CA252649 to H.A.P.; P50 CA168504 to H.A.P.; K12 CA076930 to J.H.; T32 HL007093 to J.H.; T32 CA247815 to K.A.C.), CDMRP grant W81XWH-18-10406 (to P.S.N), Komen Breast Cancer Foundation Catalyst Research Grant (to H.A.P.). Scientific Computing Infrastructure was funded by ORIP Grant S10OD028685.

## Author contributions

A.-L.D. and G.H. conceived the study, designed all the experiments, and wrote the manuscript. A.-L.D. developed and implemented the Griffin framework and associated methodologies and performed all the analysis. M.K., H.L., A.E.C, C.K., A.C.H.H., K.C., A.Z. contributed to the analysis. K.S., K.A.C., H.A.P., D.G.S. provided sequencing and clinical data. J.H., R.D.P., N.D.S., M.A., J.R., Z.T.W. contributed to analysis discussions. P.P., V.A.A., P.S.N., D.M., H.A.P., D.G.S. contributed to discussions, provided guidance and interpretation of results. G.H. supervised the study. All authors reviewed and edited the manuscript.

## Competing interests

G.H., A.L.D., J.B.H., R.D.P., D.M., P.S.N., N.D.S., are inventors on a patent application (PCT/US2022/024082) entitled CELL-FREE DNA SEQUENCE DATA ANALYSIS METHOD TO EXAMINE NUCLEOSOME PROTECTION AND CHROMATIN ACCESSIBILITY submitted by Fred Hutchinson Cancer Research Center relating to the methodologies developed and applied in this manuscript. P.P. is now an employee of C2i Genomics.
