## [Peer Review File · Nature Communications]

A framework for clinical cancer subtyping from nucleosome profiling of cell-free DNAREVIEWER COMMENTS

Reviewer #1 (Remarks to the Author): Expert in cell-free DNA and cancer genomics

Anna-Lisa Doebley, Gavin Ha and colleagues have described a computational approach for analysis of open chromatin sites from cfDNA whole genome sequencing data to distinguish cfDNA between cancer patients and healthy individuals, identify cancer type and potentially identify cancer subtype. This work relies on publically available WGS data and is an interesting informatics step forward but its claims of novelty are limited or underdeveloped. My specific concerns are below:

1. It is unclear what the effect size and direction of GC bias correction are across this dataset, and whether those lead to better overall performance of Griffin for detection of cancer, cancer type or subtype. There are multiple challenges here. First, the GC bias assessment is somewhat complicated because GC content itself is non-uniform across TF binding sites, and this is further exacerbated by sample-specific analytical (not described or evaluated). Second, how much difference does there have to be between GC content of difference fragments to generate a significant effect in coverage? Third, does it really matter when average coverage is 0.1x, the proposed application space for Griffin?
2. Novelty is limited for each of the proposed applications since they have been described recently. Detection of cancer by integrating multiple sites across the genome was shown in Cristiano et al. Nature 2019 and at lower depths by Budhraj et al. medRxiv 2021 earlier this year. Detection of cancer subtypes through integration of multiple features was shown by Ulz et al. Nature Genetics 2016, tissue-specific contribution in cfDNA from cancer patients using genomewide integration across the breadth of loci by Snyder et al. Cell 2016 and Markus et al. Science Translational Medicine 2021, and subtype prediction by TFBS analysis by Ulz et al. Nature Communications 2019.
3. Subtype prediction using 0.1x or similar sequencing depth is interesting but this application seems very limited in scope to be widely relevant since it only really works when tumor fraction is at least 10%. Median tumor fraction at presentation in ER+ breast cancer was reported at ~2.4% - this is a cohort of patients generally selected for refractory advanced cases (Dawson et al. NEJM 2013).
4. Moreover, prediction of ER+ alone even at presentation is unlikely to be enough to yield clinical relevance and for metastatic breast cancer, HER2 assessment is at least an additional need. The extension and wide applicability of these findings beyond ER+ status seem very challenging since they rely on dramatic shifts in chromatin accessibility to achieve sensitivity from low pass WGS data.
5. The clinical examples identified in Fig 4def, seem speculative, especially in the absence of known ER status for any of the subclones in Fig 4f. It is strange that cfDNA would bear more resemblance in the quoted examples to the primary (presumably profiled years ago) compared to the metastatic ER status - this is a departure from prior findings of comparisons in mutation profiles. For the serial analysis of subclones, while longitudinal changes cellular prevalence of a particular clonal cluster seems to change in step with ER probability from cfDNA, this seems to have little to do with relevant cellular prevalence of the cluster. One would expect ER probability to be higher when that clonal cluster constitutes the highest share of tumor content in plasma.

Reviewer #2 (Remarks to the Author): Expert in computational cancer genomics and cell-free DNA

The authors present a computational framework for analysing nucleosome positioning at transcription factor binding sites (TFBS) in cell-free DNA from cancer patients. The work extends previous work (Ulz et al. 2016 and 2018) showing that active TFBS can be inferred from the coverage profiles of cfDNA to predict presence of cancer. The main innovation of this paper is an improved analytical framework (e.g. correcting for GC content) and a proof-of-concept example that this approach could be used to infer subtypes of cancer (here ER+/ER- subtype in breast cancer). Overall, this is an interesting study that highlights the utility of this analytical approach and equips the field with new tools to study cfDNA. The paper is also well structured and written. However, the study would benefit from additional validation of predictive accuracy in external cohorts and analysis of detection limits. Please see my main and minor concerns listed below.

Main concerns

1) Evaluation of accuracy in independent cohorts

Multiple models are developed in this study. Models that predict presence/absence of cancer, tissue of origin, and subtypes of breast cancer. However, it appears all models are developed and tested using data from the same studies/cohorts. cfDNA samples could potentially be affected by study specific biases and technical artefacts. It is important that the authors test the robustness of their models when applied to independent cohorts from other studies. E.g. the cancer detection model developed on p. 8 using data from [38] could be tested on the breast cancer and healthy subjects data used in the later part of the manuscript. Similarly, the authors could test their tissue-of-origin model developed from [38] on the breast cancer dataset as well as some of the many other publicly available lpWGS cfDNA datasets. Finally, all models (detection, tissue-of-origin, cancer subtype) and benchmarks (code / data when possible) should be made available via GitHub for external use and validation in other studies.

2) Limit of detection

The authors have not formally tested the limit of detection of their models (detection, tissue-of-origin, breast cancer subtypes). This is important, as a model that can detect cancer only when samples have >5-10% ctDNA fraction is not really useful. The authors should test the limit of detection of all their models, for example using the strategies used in the ichorCNA study (Adalsteinsson et al. 2017).

3) Blood and cancer specific TFs not systematically analyzed

In the analysis presented on page 7 and 8 the authors show one example of a cancer-specific TF (GRHL2) and one example of a blood-specific TF (LYL1). The authors should more systematically define blood and cancer/epithelial specific TFs (e.g. using Encode/Roadmap epigenomics consortium data) and show that these correlation patterns are preserved across multiple TFs and not just isolated instances. Moreover, the authors show RMSE for all TFs in Fig 2f. Here they should also show the correlation coefficient distribution separately for cancer and blood-specific TFs.

4) Clonal clusters / Figure 4f

Fig. 4f needs to be further explained. What are the clonal clusters composed of (which mutations / CNVs)? Are these mutations associated with ER+ tumors? It is also not clear if the correlations between ER+ probability and individual clusters are stronger than would be expected by chance. The authors could test this by shuffling ER+/ER- patient labels in the training data, creating a "random" ER+ model, and showing that the correlations with clonal clusters is stronger than the random model.

Minor points

5) Method vs. Framework

I agree with the authors that they should refer to their study as presenting a 'framework' rather than a 'method'. However, the manuscript does at certain places refer to this as a 'method', e.g. in abstract. The authors should be consistent.

6) Prediction of tissue-of-origin

In Figure 3d the authors show the performance of their tissue-of-origin model. It is not entirely clear how good this performance is (Overall AUC=0.6). The authors could compare the performance with a naive model that predicts tissue-of-origin with a probability proportional to the proportion of samples from the given tissue.

7) TNBC samples

On page 12 the authors refer to 6 TNBC (triple-negative?) patients. If these are triple-negative patients, why do some of the patients have ER+ lesions (e.g. MBC_1405)?

8) Fourier transform

The authors should show data demonstrating that the Fourier transform features provide a meaningful improvement to the model (Figure 1).

9) GC-correction

Multiple existing methods have been published that perform GC-correction on genomics data (e.g. CNVkit, deepTools). Why not just use these existing tools for GC correction? It would be useful if the authors could demonstrate (using data, e.g. Figure 2) that their GC-correction approach is better tailored for cfDNA than conventional approaches.

10) Fig 4g?

The main text mentions Fig. 4g, I assume it is a typo (4f?).

Reviewer #3 (Remarks to the Author): Expert in cell-free DNA and breast cancer genomics

In this study, Doebley and colleagues present a method which exploits non-random fragmentation of circulating DNA and related sequencing coverage patterns and examines its utility towards detecting cancer and classifying tumour types in published datasets. The authors build on previous ctDNA studies and concepts in the fragmentomics field for this purpose. I commend the authors for providing a clear and detailed methods section that allows for a thorough understanding of their work.

To date, the majority of ctDNA applications have been centred around characterising genomic changes but the ability to couple cfDNA analysis with understanding transcriptional patterns and phenotypic assessment highlights the importance of this work. Despite the importance of the work there are several technical issues that have been identified which could impact the results and usefulness of the methodology, as detailed below:

1. GC content and mappability are two well-studied technical biases that affect sequencing coverage across the genome. However, many cancers have genomes that are highly copy number aberrated (CNA) resulting in a bias that is biological in nature. For example, focal copy number amplifications at sites increase the number of fragments quantified leading to the inference that the sites are inaccessible. Therefore, it is crucial to correct for the coverage bias that is due to CNA in order to minimise confounding of the coverage patterns at regions of interest. The authors do not address this bias anywhere in the manuscript when it is highly relevant especially in CNA-driven breast cancer.
2. The authors employ a method to correct the effect of GC content but have elected not to correct for mappability. Previous studies that use ultra-low pass sequencing have shown that joint correction for both mappability and GC provide great reduction in the noise of the coverage profile (Scheinin et al. 2014). The authors opt instead to restrict the analysis to regions within a narrow range of unique mappability (>0.95) and reduce the list of transcription factors analysed to those with more than 10,000 highly mappable sites on autosomes. (i) The authors should provide details on how these cutoffs were derived. (ii) Quantify any differences in number and type of regions that remain with different mappability thresholds applied. (iii) Quantify and discuss the effect on the accuracy/AUC estimates in cancer detection & tumour type classification.
3. The manuscript stresses the novelty of the GC bias correction implemented in Griffin which accounts for the relationship between fragment coverage and the GC content of the template DNA fragments which in circulating DNA is affected by the size of the DNA fragments. This relationship was first discovered and comprehensively investigated by Benjamini and Speed who described and implemented a single-position GC bias correction model (Benjamini & Speed 2012) that accounts for mappability and fragment size specific GC content. The manuscript states that this 2012 framework "assumes that all fragments have the same length". However, The Benjamini & Speed model has the flexibility to account for different fragment size ranges as well as fragment motif differences and these concepts have been used in both RNA-seq data (Love, Hogenesch & Irizarry 2016) and plasma sequencing data for non-invasive prenatal testing (Chandrananda et al. 2014). The authors should give due credit to this seminal work by Benjamini & Speed and clearly state how Griffin builds on this work for application to ctDNA.

4. Furthermore, it appears that the authors perform the assessment of Griffin before and after GC correction using all fragment lengths present in the WGS data. However, all other TFBS analyses and nucleosome profiling were carried out using a very narrow range of fragment lengths (100 – 200 bp). The authors should present data to justify this decision? Why weren't the full fragment length distribution used if the GC bias correction accounts for varied fragment sizes?

5. The limitations of machine learning are well understood and chief among these is "over-fitting" of the model to training data. This study uses a bootstrap framework and sampling with replacement. In addition, the datasets used have multiple samples from the same patient (even if they are not split across training & testing sets). These factors are strong indicators that the models trained could be prone to over-fitting. It is critical to test the models on validation data that is independent of training samples.

6. The authors always present the AUC and accuracy from the bootstrap iterations as a mean and 95% confidence interval. It would instead be useful for a reader to see the actual values of these estimates as boxplots to show the full range of accuracy & AUC values including outliers.

7. The authors demonstrate that they achieved an AUC of 0.88 for cancer detection when the cfDNA data was downsampled to 0.1x coverage. It would be important to understand how this differed across different stages (i.e. stage I vs stage IV disease).

8. The approach was able to predict the tissue of origin in 60% of samples but this was not performed on the downsampled data, despite a major focus of the manuscript being on the use of Griffin to detect cancer accurately from ULP-WGS. This should be clarified in the manuscript and the effect of downsampling the data on the TOO predictions should be shown.

9. In the ER subtyping, the authors show that seven patients who demonstrated ER loss between primary and metastatic disease, were predicted to still be ER+ by the cfDNA subtyping. Without an example of a case or cases with multiregional tumour sampling or single cell sequencing to confirm evidence of heterogeneity in ER expression at the time of the cfDNA sampling, it is difficult to know whether or not the cfDNA predictions are accurate. The relationship between the cfDNA predictions and the subclonal populations are also unsubstantiated in Figure 4f. For example, MBC_1405 is described as having an ER+ primary and one ER+ metastatic biopsy (25% expression by IHC) but all five timepoints for this case were predicted to be ER- which raises concern about the robustness of the predictions.

Reviewer #4 (Remarks to the Author): Expert in epigenomics, ChIP-seq and ATAC-seq

In this manuscript, Doebley et al develop and evaluate Griffin, a tool to identify cancer subtypes based on inferred nucleosome profiles from cell-free DNA data. Their pipeline uses fragment size selection and GC-correction to improve performance. The latter particularly important given the high GC bias of regulatory features, such as TF binding sites. The authors provide appropriate evidence supporting that their method can accurately identify cancer subtypes based on the inferred nucleosome protection profiles patterns in cfDNA sequencing data, including ultra low-pass whole genome sequencing. Overall, the manuscript is sound, but would benefit from additional figures/analyses detailing the results, including a more thorough characterization of the discriminatory features learned by Griffin models.

Major issues

* The authors developed a GC bias correction approach that differs from existing methods by accounting for fragment lengths. Given the novelty of this approach, the authors should include supplementary plots detailing their methodology and intermediary steps, including showing the normalized GC bias distribution for one sample and the corresponding values after correction, and a summary heatmap or boxplot showing the RMSE values for all features before and after GC correction in one of the cfDNA datasets. In addition, it would be informative to show which GC categories are enriched to overlap the features used as input in the Griffin models. This would reinforce the importance of performing GC bias correction step for this type of analyses.

* From a biological standpoint, the study would benefit from a more in-depth characterization of the models learned by Griffin, including which biological features drive the predictions. For example, the authors could include a summary figure reporting which features are more informative for each cancer type in the different datasets analyzed (e.g. based on the logistic regression coefficients in the final model). This type of exploration will make the underlying Griffin models more interpretable and also has the potential to uncover novel cancer biology.

* The selection of the ER+/- specific sites in the TCGA ATAC-seq data is overly permissive and a non-parametric test does not correspond to the most appropriate statistical framework, in my opinion. An $\text{abs}(\log_2 \text{ fold-change})$ of 0.5 seems very lenient to call subtype-specificity. It is not clear if the authors accounted for technical covariates were in the differential analyses (sample sequencing depth, etc). The authors should perform differential accessibility analyses using tools designed for count-based sequencing data, that account for technical factors (e.g. DESeq2). By using a more appropriate statistical framework and stringent cutoffs, the authors should be able to identify subtype-specific accessible regions with higher confidence, which will likely reflect in improved performance for Griffin to separate between ER subtypes (particularly in the low Tx samples).

Minor issues

* The cfDNA dataset used by the authors for the analyses in Fig. 3a-b and Sup. Fig 3 a-d is well-balanced when considering healthy controls vs cancer patients (n=215 vs n=208). However, the dataset becomes highly imbalanced when evaluating performance in the specific subtypes (e.g. n=12 lung cancer patients). This can affect the ROC-AUCs interpretation. The authors should include precision-recall curves and AUCs for these comparisons, which are more robust to class imbalance. The same also applies for Fig 4c and Sup. Fig 5.

* The authors should include diagnostic plots of the differential analyses (e.g. M-A plots) to allow better evaluation of the data normalization and differential accessibility results.

* If the authors wish to make more informative comparison with the Ulz pipeline, they may consider running it using the same fragment sizes as used by Griffin. This will allow to determine how much the GC content bias correction and the usage of smaller fragments sizes each contribute for the improved performance in Griffin.

* For future iterations of the Griffin pipeline, I believe the sensitivity could be further improved by including features that are more specific to cancer subtypes of interest (e.g. TFBS from ChIP-seq experiments comparing ER+ to ER- tumors).

* It is not appropriate to call the differentially accessible regions in the ER+/- as "subtype-specific". This would imply that they are only accessible in one but not in the other subtype, which does not correspond to the differential analyses performed by the authors (particularly considering the 0.5 \log_2 fold-change cutoff currently used).

* The authors should refrain from using subjective descriptors to their results, such as "excellent" (line 7) or "great" (line 170).

* I believe a reference to panel 4e is missing from line 244 in the main text.

We thank the reviewers for their thoughtful reading of our manuscript and for providing excellent suggestions. We believe the manuscript has greatly benefitted from these comments. Below, we have included a brief summary of the major analysis revisions, followed by point-by-point responses to reviewer comments.

Summary of key revision analyses

We have performed four key analyses that directly address comments from multiple reviewers, in addition to addressing each point raised. These analyses are the following:

- 1) Validation cohorts for cancer detection and ER subtyping in breast cancer. Reviewers #2 and #3 suggested validating the trained models using Griffin features in independent cohorts for each application. We now provide results in which we trained and tested a model for cancer detection using the LUCAS training and validation cohorts (Mathios et al. Nat Commun, 2021; PMID: 34417454). For ER subtyping, we compiled three cohorts consisting of new ULP-WGS cfDNA data and data obtained from published studies. See Responses to Reviewer #2 (comment 1) and Reviewer #3 (comment 5) for details. These results are presented in Fig. 3c-d, Fig. 4e, Supplementary Fig. 6, Supplementary Fig. 13, Supplementary Data 7, and Supplementary Data 14.
- 2) Inclusion of additional ER status from multiple tumor biopsies for patients with longitudinal plasma samples. We agree with the concerns raised by all reviewers that the comparison between ER subtype predictions and clonal analysis was speculative. We have now removed the clonal analysis from the study. Obtaining single-cell or multi-regional biopsy sequencing data, or even synchronous tissue and plasma, is challenging and not possible in this retrospective analysis and we believe is beyond the scope of this study. We have highlighted these limitations and removed conclusions regarding clonal dynamics involving ER subtypes. In its place, we obtained additional pathology information for multiple biopsies in some of the patients during their metastatic disease course. In key examples, we show that the ER subtype prediction was concordant with or partially explained by the ER status (by IHC) of a recent biopsy. These results are presented in Figure 4f-h, Supplementary Fig. 14, Supplementary Data 15.
- 3) Systematic comparisons between various Griffin configurations and feature. All reviewers provided suggestions to systematically evaluate the benefits of various features of Griffin for both cancer detection and ER subtyping applications. We now present performance evaluations for the following configurations/features of Griffin.
 - a) Fragment size ranges: short fragments (35-150 bp), nucleosome-sized fragments (100-200 bp), all fragments (35-500bp)
 - b) Explicit copy number correction
 - c) Inclusion/exclusion of Fourier transform amplitude feature
 - d) GC and mappability correction configurations: no GC correction, GC & mappability correction, GC correction using single fragment length (i.e. deepTools using 165 bp), GC correction accounting for multi-length fragments (current Griffin).
 - e) Previous comparisons in original submission: DNase hypersensitivity I sites, comparison to Ulz et al. (Nat Commun, 2019; PMID: 31604930) methodThese results are presented in Fig. 3b, Supplementary Fig. 7a and Supplementary Fig. 10.
- 4) Identification of ER subtype-specific differentially accessible sites from ATAC-seq data using DESeq2. Reviewer #4 suggested using an established computational tool for identifying open chromatin sites differentially accessible in each ER subtype. We now performed this analysis

using DESeq2 and show the performance evaluation for various DESeq2 thresholds, including log₂-fold-change and q-value. These results are presented in Figure 4a-b, Supplementary Fig. 11, Supplementary Data 9.

Due to changes in the analysis, as suggested by the reviewers, we would like to note some key differences compared to the previous version of the manuscript.

- We evaluate ER performance using the first available plasma sample for each patient, and the ER label was determined as the most recent tumor biopsy to that blood draw. After receiving new data for ER IHC status of multiple biopsies for some of the MBC patients in this study, the ER label used for classification performance evaluation have changed for four patients (MBC_331, MBC_1413, MBC_1405, MBC_1399).
- For the TFBS analysis, we now use different number of transcription factors (TFs) for each analysis. In the assessment and comparisons of GC and mappability correction results, we used 377 factors. This differs from the original 388 because the new mappability filtering strategy led to more TFs having > 10k TFBSs. For classification applications (cancer detection & ER subtyping), we used 270 TFs because they contained at least 30,000 sites, which we had determined to provide the optimal performance in ULP-WGS. However, note that Griffin can use any number of sites for each TF. These decisions are described in the methods.
- The reviewers provided insightful comments and concerns for the tissue-of-origin analysis. However, after careful consideration, we have removed the tissue-of-origin analysis from the manuscript for several reasons. First, the focus of this manuscript is not to determine localization of the cancer. Second, Griffin's tissue-of-origin analysis is too preliminary and requires further investigation that is beyond the scope of this manuscript. Finally, the theme and flow of the manuscript is now more tailored and focused on the innovative aspects of the study – breast cancer subtyping.

We provide responses to specific reviewer comments below.

Reviewer comments are shown in *light italics*, our responses are shown in **bold**, and excerpts from the revised manuscript are indented and in ***bold italics***.

Reviewer #1 (Remarks to the Author): Expert in cell-free DNA and cancer genomics

Anna-Lisa Doebley, Gavin Ha and colleagues have described a computational approach for analysis of open chromatin sites from cfDNA whole genome sequencing data to distinguish cfDNA between cancer patients and healthy individuals, identify cancer type and potentially identify cancer subtype. This work relies on publically available WGS data and is an interesting informatics step forward but its claims of novelty are limited or underdeveloped. My specific concerns are below:

1. It is unclear what the effect size and direction of GC bias correction are across this dataset, and whether those lead to better overall performance of Griffin for detection of cancer, cancer type or subtype. There are multiple challenges here. First, the GC bias assessment is somewhat complicated because GC content itself is non-uniform across TF binding sites, and this is further exacerbated by sample-specific analytical (not described or evaluated). Second, how much difference does there have to be between GC content of difference fragments to generate a significant effect in coverage? Third, does it really matter when average coverage is 0.1x, the proposed application space for Griffin?

We agree that the GC content is non-uniform across TF binding sites and that there are sample-specific technical biases. These are key concerns that the GC correction procedure addresses. We illustrated these challenges in Figure 2a-c by presenting the variability in GC content across 1000 GRHL2 sites (Figure 2a) and the differences in GC bias between an MBC and a healthy donor (HD) sample (Figure 2b). Griffin performs GC correction on a fragment level, thus, normalizing nucleosome coverage at each individual site, prior to aggregation into composite coverage profiles. We have indicated this in Figure 1b (step 2), and we have clarified this point in the Results (line 98).

To further demonstrate this more systematically for all TFs (besides GRHL2), we now show the variability in GC content between the binding sites and the flanking regions, including a systematic increase in GC content at the binding site (center, median 0.51 GC content) compared to the flanking region (median 0.47 GC content) for the majority (344/377) of TFs and their binding sites. We also note that the reduction in central coverage variability (as measured by RMSE) afforded by GC correction was more pronounced when there was a larger difference in GC content between the binding site and flanking regions.

“At open chromatin regions, especially at TFBSs, GC-content is non-uniform between the binding site and flanking regions, which leads to GC-related coverage biases (Fig. 2a, Supplementary Fig. 1b,c, Supplementary Data 1).⁴⁶”
(Results, lines 114-117)

See also Supplementary Fig. 1b-d and Supplementary Data 1.

To better illustrate the potential effect of GC content differences on fragment coverage, Figures 2a and 2b highlight an example: the coverage bias increases by ~6% even for fragments with 0.03 GC content difference (i.e. 0.45 to 0.48 GC). We have added a description of this example to the Figure legend text to help readers better interpret Figure 2b.

“The GC contents at the site center and flanking regions for GRHL2 are noted with dashed lines at the same GC values as in (a). To illustrate this, the coverage bias values for 160bp fragments are labeled for the MBC_315 (orange dot) and healthy (blue dot) samples. The coverage bias in this MBC sample is higher for the GC value at the central site (2.11) than that for the flanking region (1.99). This difference in bias is less pronounced in the healthy donor (1.90 central, 1.96 flanking). This means that for GRHL2, we expect to see increased coverage at the site center, relative to the flanking regions, due to GC bias and that this will be more pronounced in the MBC sample.”
(Fig.2 legend, lines 1023-1030)

Figure 2d-f illustrate the advantages of GC correction for ULP-WGS (0.1x coverage) data, highlighting improved central coverages and correlation with tumor fraction for an exemplar blood-specific TF (LYL1) across 191 MBC samples (Figure 2d-e). Across all TFs, we see decreased variability in the MBC samples for the majority of factors as shown by the decreased RMSE (when comparing central coverage and tumor fraction) following GC correction (Figure 2f). We now also show systematic performance comparisons between GC correction and no GC correction for cancer detection and ER status prediction. For cancer detection using ULP-WGS, we achieved an

overall improved performance with GC correction (0.89 AUC) compared to no GC correction (0.85 AUC). For ER subtyping (ULP-WGS), we achieved slightly improved performance for GC correction (0.892 vs. 0.871 AUC).

“Next, when omitting GC correction, we also observed decreased overall performance for 1-2X WGS (AUC=0.83, Supplementary Fig. 7a) and ULP-WGS (AUC=0.85) and for all disease stages (Fig. 3b).”

(Results, lines 202-205)

See also Figure 3b, Supplementary Fig. 7a

2. Novelty is limited for each of the proposed applications since they have been described recently. Detection of cancer by integrating multiple sites across the genome was shown in Cristiano et al. Nature 2019 and at lower depths by Budhreja et al. medRxiv 2021 earlier this year. Detection of cancer subtypes through integration of multiple features was shown by Ulz et al. Nature Genetics 2016, tissue-specific contribution in cfDNA from cancer patients using genomewide integration across the breadth of loci by Snyder et al. Cell 2016 and Markus et al. Science Translational Medicine 2021, and subtype prediction by TFBS analysis by Ulz et al. Nature Communications 2019.

We appreciate the reviewer’s concerns. We agree that using Griffin for cancer detection itself is not a novel application. As a demonstration of the Griffin framework, we illustrate its ability to perform cancer detection with similar performance as previous methods on the same dataset. Moreover, we observed good performance even for low-coverage WGS data, which is more challenging and less addressed in the literature. Thank you for making us aware of the medRxiv preprint article – we now cite this in the manuscript. We have also included a statement in the Results to emphasize that the application of Griffin for cancer detection is not novel.

“Overall, while cancer detection has been demonstrated from nucleosome profiling analysis in ctDNA^{38,43-45}, we show that Griffin may also be applied in this setting.”

(Results, line 225-227)

Also, we now present a systematic comparison of the various features for cancer detection and highlight some key analytical considerations in response to comments by the reviewers (Supplementary Fig. 7a).

We agree that the detection of cancer subtypes was first shown by Ulz et al. (Nat Commun, 2019, PMID: 31604930), using ctDNA and we have made this clear in the manuscript that it has inspired the current work. However, in prostate cancer, Ulz et al. only demonstrate subtyping in six patients. We have highlighted the advances of Griffin over the Ulz method, including GC correction, use of open chromatin sites (from ATAC-seq), and application to ULP-WGS (0.1x). Furthermore, we demonstrate superior performance of Griffin over the Ulz method for both cancer detection (0.939 vs. 0.818 AUC) and ER subtype classification when using open chromatin sites (0.892 vs. 0.551) and TFBS (0.789 vs. 0.551). The ER subtyping analysis included more than 139 patients with ER IHC data from the tumor. Finally, to our knowledge, ER subtyping in breast cancer from nucleosome profiling of ctDNA has not been shown before and we believe this is an original

contribution to the field. We have now included the Markus *et al.* STM, 2021 reference in the manuscript as suggested by the reviewer.

See Supplementary Fig. 7a and Supplementary Fig. 10.

*3. Subtype prediction using 0.1x or similar sequencing depth is interesting but this application seems very limited in scope to be widely relevant since it only really works when tumor fraction is at least 10%. Median tumor fraction at presentation in ER+ breast cancer was reported at ~2.4% - this is a cohort of patients generally selected for refractory advanced cases (Dawson *et al.* NEJM 2013).*

We appreciate the reviewer's concern regarding the challenge and limitation of low tumor fraction when predicting subtypes. While the performance of ER subtyping from ULP-WGS (0.1x) data using Griffin is higher in samples with > 10% tumor fraction (≥ 0.92 AUC), we now show that for samples with tumor fractions between 5-10%, the ER classification performance is 0.75 AUC (MBC cohort) and 0.90 AUC (validation cohort)

See updated Fig. 4d-e, Supplementary Fig. 13.

Moreover, our study focused on metastatic breast cancer where we have previously reported as high as 34.5% of MBC patients with > 10% tumor fraction (Adalsteinsson *et al.*, Nat Commun, 2017; PMID: 29109393). We do not suggest being able to perform subtyping in patients at presentation of primary localized disease. We have added this limitation to the Discussion.

“A limitation of the binary ER classification (ER+ or ER-) is the decreased accuracy for samples with lower tumor fraction (< 10%) and a 5% limit for accurate prediction, suggesting that it may be challenging to use Griffin for early-stage and minimal residual disease settings. However, in MBC, previous reports have suggested that up to 34% of MBC patients may have at least 10% tumor fraction in plasma¹⁰, which highlights potential utility for this disease stage.”

(Discussion, lines 341-345)

Application of Griffin for ER subtyping is a first demonstration and step towards the goal of non-invasive subtyping in breast cancer. However, additional studies are needed to improve the approach for application in earlier stage cancers or plasma samples with lower ctDNA content. We emphasize this in the Discussion.

“It may be possible to improve performance of ER subtyping for lower tumor fraction samples with additional sequencing depth, using TFBSs identified directly from ER+/- tumors, or joint analysis of multiple cfDNA timepoints from the same patient.”

(Discussion, lines 346-348)

4. Moreover, prediction of ER+ alone even at presentation is unlikely to be enough to yield clinical relevance and for metastatic breast cancer, HER2 assessment is at least an additional need. The extension and wide applicability of these findings beyond ER+ status seem very challenging since they rely on dramatic shifts in chromatin accessibility to achieve sensitivity from low pass WGS data.

We agree that the assessment of HER2 is important for metastatic breast cancer. The dramatic shifts in chromatin accessibility are pronounced when considering ER status, as our data demonstrates (Figure 4b). Identifying chromatin accessibility differences for HER2 status may be challenging; however, we discussed assessing ERBB2 copy number amplification as has been shown previously from ctDNA. We have added references to literature that have applied ERBB2 copy number analysis in ctDNA (Bujak et al. PLOS Med, 2020, PMID:33001984 ; Guan et al. Breast, 2020, PMID: 31927339) in lines 367-370.

5. The clinical examples identified in Fig 4def, seem speculative, especially in the absence of known ER status for any of the subclones in Fig 4f. It is strange that cfDNA would bear more resemblance in the quoted examples to the primary (presumably profiled years ago) compared to the metastatic ER status - this is a departure from prior findings of comparisons in mutation profiles. For the serial analysis of subclones, while longitudinal changes cellular prevalence of a particular clonal cluster seems to change in step with ER probability from cfDNA, this seems to have little to do with relevant cellular prevalence of the cluster. One would expect ER probability to be higher when that clonal cluster constitutes the highest share of tumor content in plasma.

We thank the reviewers for raising concerns regarding the comparison of the ER prediction. We have updated the analysis to highlight that for patients with no ER subtype switches, the prediction performance is high (new Figure 4c,f). This applies to the majority of patients (92/101). However, for the nine patients with ER loss, we also agree with the reviewer that it was unexpected that we observed an enrichment of ER+ predictions for ER loss patients ($p=3.7 \times 10^{-4}$). We now present additional metastatic tumor ER immunohistochemistry (IHC) information for multiple biopsies in some of these patients. For these ER loss patients, we noted instances where both ER+ and ER- metastatic tumors were biopsied. The presence of ER+ and ER- tumors in the same patient, and sometimes within brief time frames (e.g. MBC_1405), suggests that the chromatin accessibility profiles predicted from ctDNA might reflect mixed or heterogeneous subtypes (Supplementary Fig. 14, Supplementary Data 15). However, future work to investigate these dynamics of intra-patient subtype heterogeneity is needed and likely beyond the scope of this manuscript (since multi-regional biopsies, single-cell profiling, and multiple synchronous biopsies and plasma not available). We have revised the text to minimize conclusions of subtype heterogeneity and emphasized the challenge in predicting potential subtype mixtures.

Furthermore, we agree with the reviewers that the comparison to clonality results were speculative and that confirming the presence of ER subtype clonality changes over time will not be possible with this patient cohort and likely beyond the scope of this study. After careful consideration, we have elected to remove this analysis from the manuscript. In its place, we now present examples of patients with longitudinal plasma and available ER status from multiple metastatic tumor biopsies (Figure 4h, Supplementary Fig. 14). We show instances of ctDNA ER prediction changes that might be explained, in part, by the ER status changes from additional biopsies taken during the disease course.

See Results (lines 292-304), Discussion (lines 351-360), Figure 4c,f,g,h, Supplementary Fig. 14, Supplementary Data 15.

See also Responses to Reviewer #2, Comment #4 and Reviewer #3, Comment #9.

Reviewer #2 (Remarks to the Author): Expert in computational cancer genomics and cell-free DNA

The authors present a computational framework for analysing nucleosome positioning at transcription factor binding sites (TFBS) in cell-free DNA from cancer patients. The work extends previous work (Ulz et al. 2016 and 2018) showing that active TFBS can be inferred from the coverage profiles of cfDNA to predict presence of cancer. The main innovation of this paper is an improved analytical framework (e.g. correcting for GC content) and a proof-of-concept example that this approach could be used to infer subtypes of cancer (here ER+/ER- subtype in breast cancer). Overall, this is an interesting study that highlights the utility of this analytical approach and equips the field with new tools to study cfDNA. The paper is also well structured and written. However, the study would benefit from additional validation of predictive accuracy in external cohorts and analysis of detection limits. Please see my main and minor concerns listed below.

Main concerns

1. Evaluation of accuracy in independent cohorts

Multiple models are developed in this study. Models that predict presence/absence of cancer, tissue of origin, and subtypes of breast cancer. However, it appears all models are developed and tested using data from the same studies/cohorts. cfDNA samples could potentially be affected by study specific biases and technical artefacts. It is important that the authors test the robustness of their models when applied to independent cohorts from other studies. E.g. the cancer detection model developed on p. 8 using data from [38] could be tested on the breast cancer and healthy subjects data used in the later part of the manuscript. Similarly, the authors could test their tissue-of-origin model developed from [38] on the breast cancer dataset as well as some of the many other publicly available lpWGS cfDNA datasets. Finally, all models (detection, tissue-of-origin, cancer subtype) and benchmarks (code / data when possible) should be made available via GitHub for external use and validation in other studies.

We have now analyzed multiple separate independent cohorts to validate the models and confirm their performance.

For cancer detection, we obtained data from EGA published in Mathios et al. Nat Commun, 2021 (PMID: 34417454) consisting of a training and validation dataset from the LUCAS cohort. The training cohort included plasma sequencing from 129 lung cancer patients and 158 healthy donors; the validation cohort included 46 lung cancer patients and 385 healthy donors. Importantly, the healthy donor dataset was generated similarly to the cancer patient data and provided an unbiased evaluation. From our assessment of the DELFI (Cristiano et al. Nature, 2019; PMID: 31142840) and the LUCAS cohorts, we noted that there were major batch effects, which prevented combining these data for evaluating the model (Supplementary Fig. 8).

The cross-validation performance for cancer detection on the LUCAS training dataset was 0.76 AUC, and the performance was further categorized by disease stage and tumor fraction (AUC 0.57-0.79). Another advantage of the LUCAS dataset was the availability of a validation cohort for direct evaluation of models trained on the LUCAS training cohort. For the validation cohort, we had an overall performance of 0.86 AUC (0.83 for Stage I, 0.86 for Stage II, and 1.00 for Stages III/IV). We

also evaluated performance for down-sampled coverages of 0.1x and observed an overall AUC of 0.69 (0.69 for Stage I, 0.65 for Stage II, and 0.81 for Stage III/IV).

See Results (lines 211-227), Methods (lines 835-868), Figure 3c-d, Supplementary Fig. 6.

For ER status prediction in breast cancer, we used three separate cohorts. (1) We analyzed a cohort of 14 patients from Ghana who had blood collected at the time of initial diagnosis (Ahuno *et al.* *npj Precision Oncology*, 2022; PMID: 34535742). ULP-WGS (0.1x) was performed on the 14 plasma samples (7 ER+, 7 ER-). We applied the model that was trained on the original MBC cohort (254 samples, 139 patients) to classify ER status in the Ghana cohort and achieved an overall performance of 79% accuracy when considering all samples, but we observed a 100% accuracy when considering only samples with ≥ 0.05 tumor fraction (n=7). (2) In a cohort of 27 patients with ER+ breast cancers who had plasma WGS performed (Bujak *et al.* *PLOS Medicine*, 2020; PMID: 33001984), we down-sampled the sequencing coverage to 0.1x. Applying our trained model, we classified ER+ in 85% of all samples and 100% when considering only samples with ≥ 0.05 tumor fraction. (3) Finally, we sequenced and analyzed samples from 30 patients (15 ER+, 15 ER-), correctly classifying ER status with 80% accuracy, including 90% accuracy for samples ≥ 0.1 tumor fraction. Overall, when combining the prediction performance for all three cohorts, we achieved an overall AUC of 0.96 (≥ 0.05 tumor fraction, n=36), which included 0.90 AUC (0.05 – 0.1 tumor fraction, n=12) and 0.98 AUC (≥ 0.1 tumor fraction, n=24).

See Results (lines 265-274), Methods (lines 928-940), Figure 4e, Supplementary Fig. 13

We thank the reviewers for your insightful comments on the tissue-of-origin analysis that prompted us to exclude it after careful consideration. We have removed the tissue-of-origin analysis from the manuscript for several reasons. First, the focus of this manuscript is not to determine localization of the cancer. Second, Griffin's tissue-of-origin analysis is too preliminary and requires further investigation that is beyond the scope of this manuscript. Finally, the theme and flow of the manuscript is now more tailored and focused on the innovative aspects of the study – breast cancer subtyping.

Code for the analyses and the trained models are included on GitHub.

Griffin pipeline:

<https://github.com/adoebley/Griffin>

Analyses, trained models, and code for regenerating manuscript figures:

https://github.com/adoebley/Griffin_analyses

2. Limit of detection

*The authors have not formally tested the limit of detection of their models (detection, tissue-of-origin, breast cancer subtypes). This is important, as a model that can detect cancer only when samples have >5-10% ctDNA fraction is not really useful. The authors should test the limit of detection of all their models, for example using the strategies used in the ichorCNA study (Adalsteinsson *et al.* 2017).*

We share the same concerns as the reviewer and agree that it is important to identify the limit of detection. We do wish to clarify that the analyses involving the 5-10% ctDNA fraction range applies

to the metastatic breast cancer (MBC) subtype prediction and not cancer detection. MBC is an advanced stage disease in which up to 73% of patients may have detectable ctDNA ($\geq 3\%$) and up to 35% have $\geq 10\%$ ctDNA (Adalsteinsson *et al.* Nat Commun, 2017; PMID: 29109393 and Stover *et al.* JCO, 2018; PMID: 29298117). Regardless, we analyzed all samples in the MBC validation cohorts and now report the performance for 0-5% ctDNA fraction (0.39 AUC) and illustrate that 5% is the likely lower limit for accurate prediction since 5-10% performed adequately (0.90 AUC) for ER status classification. For the training cohort, we did not include samples with $< 5\%$ because it did not benefit model training. See Results and Discussion:

“For samples with tumor fraction < 0.05 ($n=35$), the accuracy was 0.54 (AUC=0.39), indicating the lower limit of accurate ER classification is likely at 0.05 tumor fraction (Supplementary Fig. 13).”

(Results, lines 270-272)

“A limitation of the binary ER classification (ER+ or ER-) is the decreased accuracy for samples with lower tumor fraction ($< 10\%$) and a 5% limit for accurate prediction, suggesting that it may be challenging to use Griffin for early-stage and minimal residual disease settings.” (Discussion, lines 341-343)

See also Supplementary Fig. 13.

For cancer detection, we had previously divided the performance into ctDNA fraction ranges of 0.00, 0-5%, and $> 5\%$ (estimated by ichorCNA). Since ichorCNA has a 3% limit of detection for ULP-WGS and we use its tumor fraction estimation for our evaluation, we have now changed to the ranges of 0-3%, 3-5%, $>5\%$. In the LUCAS training cohort, we observed an AUC of 0.65 for samples with 0-3% ctDNA ($n=64$) and to 0.78 AUC for 3-5% ctDNA ($n=21$) compared to 0.91 AUC for $\geq 5\%$ ctDNA ($n=44$). In the LUCAS validation cohort, we observed an AUC of 0.87 for 0-3% ctDNA ($n=40$) and 0.81 AUC for $\geq 3\%$ ctDNA. Thus, we have identified the limit of detection to be $\sim 3\%$ tumor fraction based on the LUCAS training and validation cohorts.

“Applying the trained model from the LUCAS cohort to the LUCAS validation cohort, we achieved an AUC of 0.86 across all stages, including an AUC of 0.83 for stage I cancers (Fig. 3d, Supplementary Fig. 5d, Supplementary Data 8). The performance was 0.87 and 0.81 AUC for tumor fractions of < 0.03 and ≥ 0.03 , respectively (Supplementary Fig. 6d). For ULP-WGS coverage, the performance was 0.69 AUC for stage I and 0.69 AUC across all stages (Fig 3d, Supplementary Fig. 5d).”

(Results, lines 220-225)

See Methods (lines 835-868), Supplementary Fig. 6a-c.

3. Blood and cancer specific TFs not systematically analyzed

In the analysis presented on page 7 and 8 the authors show one example of a cancer-specific TF (GRHL2) and one example of a blood-specific TF (LYL1). The authors should more systematically define blood and cancer/epithelial specific TFs (e.g. using Encode/Roadmap epigenomics consortium data) and show that these correlation patterns are preserved across multiple TFs and not just isolated instances.

Moreover, the authors show RMSE for all TFs in Fig 2f. Here they should also show the correlation coefficient distribution separately for cancer and blood-specific TFs.

We now show the Pearson correlation coefficient distribution as boxplots comparing the three Griffin features (central coverage, mean coverage, FFT amplitude) between TFs up-regulated in blood (n=22) and in cancer (n=35) determined using the UCSC Xena browser (described in Methods). Note that while these genes may have expression associated in blood and cancers in these databases, it may not necessarily be actively binding to sites, leading accessibility in cfDNA. Nevertheless, for central coverage and mean coverage features, the values for blood factors are expected to be positivity correlated (median Pearson $r = 0.16$) with tumor fraction (see Figure 2e), and that the correlation is significantly higher after GC correction (median $r = 0.30$) (Supplementary Fig. 4g). We observed that while some known cancer-specific TFs had significant negative correlations with tumor fraction as expected, many were not correlated and did not have stronger effects following GC correction. Finding the set of TFs that are associated with immune or cancer cells, globally, is challenging and was a key reason for why we used RMSE to evaluate the improved variability of the cfDNA Griffin features

“Next, in the cfDNA samples, we systematically analyzed differentially expressed TFs between blood cells and breast cancer (Methods, Supplementary Data 4). We found that central coverage and tumor fraction were correlated for a subset of these TFs (11 of 35 cancer and 15 of 22 blood TFs, adjusted p -value < 0.05), most correlations were in the expected direction, and that these correlations increased for blood-specific TFs after GC correction (Supplementary Fig. 4a).”
(Results, lines 157-162)

See Methods (lines 680-713), Supplementary Fig. 4g, Supplementary Data 3,4.

4. Clonal clusters / Figure 4f

Fig. 4f needs to be further explained. What are the clonal clusters composed of (which mutations / CNVs)? Are these mutations associated with ER+ tumors? It is also not clear if the correlations between ER+ probability and individual clusters are stronger than would be expected by chance. The authors could test this by shuffling ER+/ER- patient labels in the training data, creating a “random” ER+ model, and showing that the correlations with clonal clusters is stronger than the random model.

The clonal clusters were determined from the analysis of somatic single nucleotide variants (SNVs) from whole exome sequencing of the same plasma samples in a previous study (Weber *et al.* *Genome Medicine*, 2021; PMID: 34016182).

We thank the reviewers for raising concerns regarding the comparison of the ER prediction probabilities with the clonal cluster data. We agree with the reviewers that the results were speculative and that confirming the presence of ER subtype clonality changes over time will not be possible with this patient cohort and likely beyond the scope of this study. After careful consideration, we have elected to remove the clonal analysis from the manuscript.

In its place, we now present additional tumor biopsy ER immunohistochemistry (IHC) information, abstracted from medical chart review, for the same patients with longitudinal plasma samples shown in Figure 4h and Supplementary Fig. 14. We show instances of ctDNA ER prediction

changes that can be explained, in part, by the ER status changes from additional biopsies taken during treatment.

See also Responses to Reviewer #1, Comment #5 and Reviewer #3, Comment #9.
See Results (lines 292-304), Figure 4h, Supplementary Fig. 14, Supplementary Data 15.

Minor points

5. Method vs. Framework

I agree with the authors that they should refer to their study as presenting a 'framework' rather than a 'method'. However, the manuscript does at certain places refer to this as a 'method', e.g. in abstract. The authors should be consistent.

We appreciate the reviewer agrees that Griffin is more of a framework. However, the GC-bias correction and normalization components do entail software code and may still be considered a tool/method. We have now replaced instances of “method” with “framework” when the overall Griffin strategy is described, but still retain “method” strictly for the GC-bias and normalization components. We hope this clarifies the terminology/semantics used in the manuscript.

6. Prediction of tissue-of-origin

In Figure 3d the authors show the performance of their tissue-of-origin model. It is not entirely clear how good this performance is (Overall AUC=0.6). The authors could compare the performance with a naive model that predicts tissue-of-origin with a probability proportional to the proportion of samples from the given tissue.

After careful consideration, we excluded this analysis from the manuscript for several reasons. First, the focus of this manuscript is not to determine localization of the cancer. Second, Griffin's analysis is too preliminary and requires further investigation that is beyond the scope of this manuscript. Finally, the theme and flow of the manuscript is now more tailored and focused on the original aspects of the study – breast cancer subtyping.

7. TNBC samples

On page 12 the authors refer to 6 TNBC (triple-negative?) patients. If these are triple-negative patients, why do some of the patients have ER+ lesions (e.g. MBC_1405)?

These 6 patients were classified as TNBC based on IHC of the initial metastatic tumor (Stover et al. JCO, 2018; PMID: 29298117) and had met the TNBC criteria for inclusion in the Cabozantinib trial. Additional metastatic biopsies were collected for some of these patients and may have been classified as ER+ with > 1% ER staining. These scenarios highlight the potential presence of ER subtype heterogeneity, which made for interesting examples to compare to longitudinal ctDNA ER prediction results. We have now added references about the TNBC trial in the methods and clarified the TNBC criteria that was applied at the time of the trial.

See Methods (lines 555-566).

8. Fourier transform

The authors should show data demonstrating that the Fourier transform features provide a meaningful improvement to the model (Figure 1).

We now provide a systematic performance comparison between inclusion and exclusion of Fourier transform amplitude feature. We evaluated the performance in the cancer detection application and found that the classification performance decreased when excluding this feature (0.75 vs. 0.94 AUC) (Supplementary Fig. 7a). For the ER subtyping classification, we noticed very marginal performance difference when comparing between inclusion (0.892 AUC) and exclusion (0.900 AUC) of this feature (Supplementary Fig. 10).

9. GC-correction

Multiple existing methods have been published that perform GC-correction on genomics data (e.g. CNVkit, deepTools). Why not just use these existing tools for GC correction? It would be useful if the authors could demonstrate (using data, e.g. Figure 2) that their GC-correction approach is better tailored for cfDNA than conventional approaches.

Griffin systematically applies GC correction to a full range of fragments sizes (35 bp to 500 bp, 1 bp increments). This approach differs from deepTools in that the latter assumes a single fragment size for its fragment-level GC correction. Moreover, the runtime using the implementation of deepTools was not practical for our needs. Because we were using ULP data, we needed to sample all positions to get as many reads as possible. Additionally, we needed to exclude unmappable and blacklisted regions to accurately estimate the GC bias. We found that (at least using our computer system) this caused deepTools to take a prohibitively long time (>1 day for ULP-WGS) while Griffin's run time was typically faster (< 1 hour). CNVkit applies GC correction to genomic bins, which is less applicable for correcting directly at specific binding sites. Griffin's GC correction approach accounting for all fragment sizes shows no performance difference over the single-fragment-length approach for cancer detection (0.939 vs. 0.939 AUC) and ER classification (0.892 vs. 0.889 AUC). However, for deeper WGS (e.g. ~20x coverage) where there is increased abundance of shorter fragments in the sequencing library, the multi-fragment-length GC correction by Griffin has a benefit for central coverage and mean coverage ctDNA features when performing nucleosome profiling on short fragments (35-100bp) (Supplementary Fig. 2).

We now provide a systematic performance comparison between GC correction configurations: no GC correction, GC correction accounting for a single-fragment length of 165bp, and GC correction accounting for multiple fragment lengths (current Griffin implementation). This comparison was performed for both cancer detection and ER status subtyping applications.

For cancer detection, see Results (lines 198-209) and Supplementary Fig. 7a.

For ER subtype prediction, see Results (lines 260-263) and Supplementary Fig. 10.

For multi-fragment length correctio comparison, see Supplementary Fig. 2 and Supplementary Data 2.

10. Fig 4g?

The main text mentions Fig. 4g, I assume it is a typo (4f?).

Figure 4 panels have changed, and the labels are now correct.

Reviewer #3 (Remarks to the Author): Expert in cell-free DNA and breast cancer genomics

In this study, Doebley and colleagues present a method which exploits non-random fragmentation of circulating DNA and related sequencing coverage patterns and examines its utility towards detecting cancer and classifying tumour types in published datasets. The authors build on previous ctDNA studies and concepts in the fragmentomics field for this purpose. I commend the authors for providing a clear and detailed methods section that allows for a thorough understanding of their work.

To date, the majority of ctDNA applications have been centred around characterising genomic changes but the ability to couple cfDNA analysis with understanding transcriptional patterns and phenotypic assessment highlights the importance of this work. Despite the importance of the work there are several technical issues that have been identified which could impact the results and usefulness of the methodology, as detailed below:

1. GC content and mappability are two well-studied technical biases that affect sequencing coverage across the genome. However, many cancers have genomes that are highly copy number aberrated (CNA) resulting in a bias that is biological in nature. For example, focal copy number amplifications at sites increase the number of fragments quantified leading to the inference that the sites are inaccessible. Therefore, it is crucial to correct for the coverage bias that is due to CNA in order to minimise confounding of the coverage patterns at regions of interest. The authors do not address this bias anywhere in the manuscript when it is highly relevant especially in CNA-driven breast cancer.

Thank you for emphasizing this critical potential confounder in coverage biases due to CNAs. Our initial expectation was that the impact of CNAs would be minimal because we analyze 1000's of sites (TFBS or open chromatin from ATAC-seq) distributed across the genome. However, we agree that CNA correction should be considered by Griffin and tested. We have now implemented an option into Griffin for site-specific copy number correction, prior to aggregating sites into composite coverage profiles. Briefly, this correction entails dividing the GC-corrected coverage values at a given site by the mean GC coverage value in the surrounding ± 50 Kbp window.

We provide a systematic performance comparison between inclusion and exclusion of copy number correction. For cancer detection, we observed very similar results between including (0.934 AUC) and excluding (0.939 AUC) CNA correction. For ER status classification, the performance for inclusion was 0.895 AUC and exclusion was 0.892 AUC; however, we noted that for tumor fraction samples $< 10\%$, exclusion of CNA correction had higher performance (0.75 vs. 0.73 AUC). Finally, because of the added computational runtime for correcting 10k to 100k sites, we decided to exclude CNA correction for analyses in the manuscript, but end-users will have the option to apply this correction.

See Methods (lines 481-489), Results (lines 198-209 for cancer detection), Results (lines 260-263 for ER subtyping), Supplementary Fig. 7, Supplementary Fig. 10

2. The authors employ a method to correct the effect of GC content but have elected not to correct for mappability. Previous studies that use ultra-low pass sequencing have shown that joint correction for

both mappability and GC provide great reduction in the noise of the coverage profile (Scheinin et al. 2014). The authors opt instead to restrict the analysis to regions within a narrow range of unique mappability (>0.95) and reduce the list of transcription factors analysed to those with more than 10,000 highly mappable sites on autosomes.

Thank you for raising these concerns about mappability correction. We have addressed these concerns in several ways, as suggested by the reviewer.

First, we implemented an explicit mappability correction procedure whereby the relationship between fragment coverage and mappability score is determined and used to correct mappability biases. However, we observed that this correction did not improve signals in ULP-WGS data (Supplementary Fig. 4a-e), likely because re-weighting individual fragments under the fragment-level normalization strategy would also include low mappability fragments which do not provide useful signals.

**“Finally, we tested the impact of mappability biases and copy number alterations (CNA) and found that explicit correction accounting for these factors did not improve RMSE values in the MBC cfDNA samples (Methods, Supplementary Fig. 4b-f, Supplementary Data 3).”
(Results, lines 169-172)**

See Methods (lines 457-480), Supplementary Fig. 4b-f.

(i) The authors should provide details on how these cutoffs were derived. (ii) Quantify any differences in number and type of regions that remain with different mappability thresholds applied.

Next, we modified the mappability filtering strategy. We agree that 0.95 cutoff was too stringent, and instead, Griffin now considers all sites when collecting coverage profiles, regardless of surrounding mappability. Then, Griffin filters out positions within profiles at known blacklist regions from ENCODE (encodeproject.org/files/ENCFF356LFX/), centromere regions, fix patches, and alternative haplotypes for hg38 (genome.ucsc.edu/cgi-bin/hgTables). Then, positions with a mappability score of zero (Umap multi-read mappability track from UCSC, Karimzadel et al. NAR, 2018; PMID: 30169659) were further filtered out. This approach removes positions with abnormal coverage while preserving sites that are nearby to these positions and is similar to the approach used in recent studies of nucleosome profiling from cfDNA (Mathios et al. Nat. Comms. 2021; PMID: 34471252; Peneder et al. Nat. Comms. 2021; PMID: 34050156) which also filtered sites using a blacklist rather than mappability cutoff. We also systematically evaluated the number of TFBSs for each TF, ranging from 1,000 – 50,000 sites, and identified that 30,000 sites provided the best cancer detection performance for ULP-WGS (AUC=0.894) compared to 1000 sites (AUC=0.874).

See Methods (lines 394-403), Supplementary Fig. 5.

(iii) Quantify and discuss the effect on the accuracy/AUC estimates in cancer detection & tumour type classification.

For cancer detection, the performance was not different when using the blacklist filter only (0.939 AUC) compared to blacklist filtering plus mappability correction (0.939 AUC). For ER classification,

the performance was also not different for blacklist filter only (0.892 AUC) compared blacklist filtering plus mappability correction (0.891 AUC). However, for individual sites, such as ZBTB16 TFBSs, we noted that mappability correction resulted in slightly lower correlation with tumor fraction ($r=0.41$ vs 0.43) and increased variability (RMSE 0.057 vs 0.051) compared to no mappability correction. Therefore, for this study, we only applied the blacklist filtering of positions within the window around sites (without thresholding); the explicit mappability correction is still an option for end-users.

See Supplementary Fig. 4a-e, Supplementary Fig. 7a and Supplementary Fig. 10

3. The manuscript stresses the novelty of the GC bias correction implemented in Griffin which accounts for the relationship between fragment coverage and the GC content of the template DNA fragments which in circulating DNA is affected by the size of the DNA fragments. This relationship was first discovered and comprehensively investigated by Benjamini and Speed who described and implemented a single-position GC bias correction model (Benjamini & Speed 2012) that accounts for mappability and fragment size specific GC content. The manuscript states that this 2012 framework “assumes that all fragments have the same length”. However, The Benjamini & Speed model has the flexibility to account for different fragment size ranges as well as fragment motif differences and these concepts have been used in both RNA-seq data (Love, Hogenesch & Irizarry 2016) and plasma sequencing data for non-invasive prenatal testing (Chandrananda et al. 2014). The authors should give due credit to this seminal work by Benjamini & Speed and clearly state how Griffin builds on this work for application to ctDNA.

We thank the reviewer for raising this concern regarding the prior description of the correction for various fragment sizes by Benjamini & Speed, 2012. We apologize for overlooking this important detail. We now appropriately attribute the GC correction of multiple fragment sizes to Benjamini and Speed, as suggested by the reviewer. To our knowledge, GC correction for multiple fragment sizes has not been applied to ctDNA analysis. Chandrananda et al. (PLoS One, 2014; PMID: 24489824) used the single-position approach for aneuploidy analysis, but they did not apply the multi-fragment-length strategy. The Griffin tool provides an implementation of the GC correction strategy that is scalable, user-friendly, and applicable to low-coverage and deep-coverage WGS of ctDNA.

Also, we now provide a comparison of GC correction between single-length and multi-length fragment strategies in ctDNA and observed for deeper WGS (e.g. ~20x coverage), where there is increased abundance of shorter fragments in the sequencing library, the multi-length correction by Griffin has a benefit for nucleosome profiling (Supplementary Fig. 2, Supplementary Data 2).

“From analysis of WGS data for 14 CRPC, two MBC, and two healthy donor samples^{10,15}, we observed stronger correlations with shorter (35-100 bp) fragments and reduced data variability when using GC correction for multiple fragment lengths, which informed the use of this correction strategy (Supplementary Fig. 2, Supplementary Data 2).”
(Results, lines 129-133)

See also Supplementary Fig. 7a, Supplementary Fig. 10 for performance comparison for cancer detection and ER subtyping prediction comparing multi-length and single-length fragment correction.

4. Furthermore, it appears that the authors perform the assessment of Griffin before and after GC correction using all fragment lengths present in the WGS data. However, all other TFBS analyses and nucleosome profiling were carried out using a very narrow range of fragment lengths (100 – 200 bp). The authors should present data to justify this decision? Why weren't the full fragment length distribution used if the GC bias correction accounts for varied fragment sizes?

Previous work has suggested that short and nucleosome sized fragments represent different types of protection in cfDNA (Snyder *et al.* Cell, 2016; PMID: 26771485); therefore, we examined these fragment size ranges separately in deep . For short fragments (35-100bp), which originate from the protection of transcription factors bound to active TFBS, we saw the expected spike in coverage at TFBS. For a blood specific factor (STAT5A), we found that this spike in coverage was negatively correlated to tumor fraction, as expected, indicating more protection in blood derived cfDNA where STAT5A is bound to the TFBS and protecting it from degradation (Supplementary Figure 2a,b). We noted stronger negative correlations for multi-length compared to single-length correction (Pearson's r -0.67 vs. -0.52), and we observed reduced signal variability (decreased RMSE) for all TFs and all three Griffin features (Supplementary Fig. 2).

However, in ULP-WGS, there were not enough total short fragments to analyze separately so we focused on nucleosome sized fragments:

“However, in ULP-WGS data from 191 MBC cfDNA samples¹⁰ with ≥ 0.1 tumor fraction, we focused on the nucleosome sized fragments (100-200bp) due to the low abundance of short fragments (<100 bp).”
(Results, lines 133-135)

We now provide a comparison of the various configurations of fragment size ranges and observed that shorter fragments (35 – 150 bp) as well as all fragments (35-500bp) led to decreased performance in low-coverage data for cancer detection (0.914 vs. 0.939 AUC). For ER status classification in the MBC cohort, the performance for short fragments (35 – 150 bp; 0.808 AUC) and all fragments (35 – 500 bp, 0.883 AUC) was lower than for nucleosomal-sized fragments (100 – 200 bp; 0.892 AUC). Regardless, the multi-length fragment GC correction was a benefit for shorter fragments in deeper WGS (Supplementary Fig. 2).

See Response to previous comment, Results (lines 133-135), Supplementary Fig. 2, Supplementary Fig. 7a, Supplementary Fig. 10.

See also Response to Reviewer #2, Comment #9.

5. The limitations of machine learning are well understood and chief among these is “over-fitting” of the model to training data. This study uses a bootstrap framework and sampling with replacement. In addition, the datasets used have multiple samples from the same patient (even if they are not split across training & testing sets). These factors are strong indicators that the models trained could be prone to over-fitting. It is critical to test the models on validation data that is independent of training samples.

We have now analyzed multiple separate independent cohorts to validate the models and confirm their performance.

For cancer detection, we obtained data from EGA published in Mathios et al. Nat Commun, 2021 (PMID: 34417454) consisting of a training and validation dataset from the LUCAS cohort. See Results (lines 211-227), Figure 3c-d, Supplementary Fig. 6c-d

For ER status prediction in breast cancer, we used three separate datasets as validation cohorts. See Results (lines 265-274), Figure 4e, Supplementary Fig. 13

See Response to Reviewer #2, Comment #1 for full details.

6. The authors always present the AUC and accuracy from the bootstrap iterations as a mean and 95% confidence interval. It would instead be useful for a reader to see the actual values of these estimates as boxplots to show the full range of accuracy & AUC values including outliers.

We now present the AUC and accuracy bootstrap iterations using a boxplot that indicate the median, interquartile range (IQR), 1.5*IQR (whiskers), and outliers. See Figure 3b, Supplementary Figs. 5-7,10,11,13

7. The authors demonstrate that they achieved an AUC of 0.88 for cancer detection when the cfDNA data was downsampled to 0.1x coverage. It would be important to understand how this differed across different stages (i.e. stage I vs stage IV disease).

The cancer detection performance for down-sampled cfDNA sequencing to 0.1x coverage is now presented across different stages in Figure 3 and Supplementary Fig. 5. Performance based on tumor fraction and cancer type categories are now shown in Supplementary Fig. 6a-c and Supplementary Fig. 6d, respectively.

8. The approach was able to predict the tissue of origin in 60% of samples but this was not performed on the downsampled data, despite a major focus of the manuscript being on the use of Griffin to detect cancer accurately from ULP-WGS. This should be clarified in the manuscript and the effect of downsampling the data on the TOO predictions should be shown.

We thank the reviewers for your insightful comments on the tissue-of-origin analysis that prompted us to exclude it after careful consideration. We have removed the tissue-of-origin analysis from the manuscript for several reasons. First, the focus of this manuscript is not to determine localization of the cancer. Second, Griffin's TOO analysis is too preliminary and requires further investigation that is beyond the scope of this manuscript. Finally, the theme and flow of the manuscript is now more tailored and focused on the innovative aspects of the study – breast cancer subtyping.

9. In the ER subtyping, the authors show that seven patients who demonstrated ER loss between primary and metastatic disease, were predicted to still be ER+ by the cfDNA subtyping. Without an example of a case or cases with multiregional tumour sampling or single cell sequencing to confirm evidence of heterogeneity in ER expression at the time of the cfDNA sampling, it is difficult to know whether or not the cfDNA predictions are accurate. The relationship between the cfDNA predictions and the subclonal populations are also unsubstantiated in Figure 4f. For example, MBC_1405 is described as

having an ER+ primary and one ER+ metastatic biopsy (25% expression by IHC) but all five timepoints for this case were predicted to be ER- which raises concern about the robustness of the predictions

We thank the reviewers for raising concerns regarding the comparison of the ER prediction probabilities with the clonal cluster data. We agree with the reviewers that the results were speculative and that confirming the presence of ER subtype clonality changes over time will not be possible with this patient cohort (since multiregional tumor sampling or single cell sequencing is not available) and likely beyond the scope of this study. After careful consideration, we have elected to remove the clonal analysis from the manuscript.

In its place, we now present additional tumor biopsy ER immunohistochemistry (IHC) information, abstracted from medical chart review, for the same patients with longitudinal plasma samples shown in Figure 4. We show instances of ctDNA ER prediction changes that can be explained, in part, by the ER status changes from additional biopsies taken during treatment. In MBC_1405, using information from additional biopsies, we were able to confirm the presence of three ER-metastases collected at later timepoints prior to and after the blood draws (Supplementary Fig. 14).

We have also updated the analysis to highlight that for patients with no ER subtype switches, the prediction performance is high (new Figure 4c,f). However, for the nine patients with ER loss, we noted instances when both ER+ and ER- metastatic tumors were biopsied. The presence of ER+ and ER- tumors in the same patient, and sometimes within brief time frames (e.g. MBC_1405) suggests that there might be mixed or heterogeneous subtypes; however, future work to investigate dynamics of intra-patient subtype heterogeneity is needed and beyond the scope of this manuscript. We have revised the text to minimize conclusions of subtype heterogeneity and emphasize the challenge in predicting potential subtype mixtures.

See Results (lines 292-304, Figure 4h, Supplementary Fig. 14

See also Responses to Reviewer #1, Comment #5 and Reviewer #2, Comment #4.

Reviewer #4 (Remarks to the Author): Expert in epigenomics, ChIP-seq and ATAC-seq

In this manuscript, Doebley et al develop and evaluate Griffin, a tool to identify cancer subtypes based on inferred nucleosome profiles from cell-free DNA data. Their pipeline uses fragment size selection and GC-correction to improve performance. The latter particularly important given the high GC bias of regulatory features, such as TF binding sites. The authors provide appropriate evidence supporting that their method can accurately identify cancer subtypes based on the inferred nucleosome protection profiles patterns in cfDNA sequencing data, including ultra low-pass whole genome sequencing. Overall, the manuscript is sound, but would benefit from additional figures/analyses detailing the results, including a more thorough characterization of the discriminatory features learned by Griffin models.

Major issues

1. The authors developed a GC bias correction approach that differs from existing methods by accounting for fragment lengths. Given the novelty of this approach, the authors should include supplementary plots detailing their methodology and intermediary steps, including showing the

normalized GC bias distribution for one sample and the corresponding values after correction, and a summary heatmap or boxplot showing the RMSE values for all features before and after GC correction in one of the cfDNA datasets. In addition, it would be informative to show which GC categories are enriched to overlap the features used as input in the Griffin models. This would reinforce the importance of performing GC bias correction step for this type of analyses.

To better illustrate the potential effect of GC content differences on fragment coverage, Figures 2a and 2b now highlight an example: the coverage bias increases by ~6% even for fragments with 0.03 GC content difference (i.e. 0.45 to 0.48 GC). We have added a description of this example to the Figure legend text to help readers better interpret Figure 2b. See Response to Reviewer #1, comment 1.

We now present data showing the RMSE for all features and healthy donor MAD before and after GC bias correction (Fig 2f and Supplementary Figs. 3b-c). Finally, to show that GC bias at the central coverage (a key feature used for downstream applications) is indeed a major confounder, we provide a systematic assessment of GC bias at all TFBSs compared to the flanking (i.e., surrounding regions) (Supplementary Fig. 1b-d). As initially illustrated in Figure 2a, we now show that the GC content at all TFBSs differs from flanking regions by a median of 0.042 GC fraction (Supplementary Fig. 1c). We also noted that the reduction in central coverage variability (as measured by RMSE) afforded by GC correction was more pronounced when there was a larger difference in GC content between the binding site and flanking regions (Supplementary Fig. 1d). This is now mentioned in the Results:

**“At open chromatin regions, especially at TFBSs, GC-content is non-uniform between the binding site and flanking regions, which leads to GC-related coverage biases (Fig. 2a, Supplementary Fig. 1b,c, Supplementary Data 1).”
(Results, lines 114-117)**

See also Supplementary Fig. 1b-d, Supplementary Fig. 3b-c, Supplementary Fig. 4d.

2. From a biological standpoint, the study would benefit from a more in-depth characterization of the models learned by Griffin, including which biological features drive the predictions. For example, the authors could include a summary figure reporting which features are more informative for each cancer type in the different datasets analyzed (e.g. based on the logistic regression coefficients in the final model). This type of exploration will make the underlying Griffin models more interpretable and also has the potential to uncover novel cancer biology.

Thank you for this suggestion to further extract information from the prediction models that may inform on the biology of the cancer types. Due to the high dimensionality of the features (3 features x 270 TFs for our cancer detection models), we now use principal components analysis to reduce the dimensionality of the data and used the top principal components that explained 80% of the variance as the features in our models. This strategy makes it more challenging to assess the original features are contributing and most associated with the predictions, as they may be split across multiple principal components. Regardless, we attempt to examine the features that contributed to the principal component with the highest coefficient in our logistic regression

model; however, most of the features with the strongest contributions to this principal component were neither blood nor cancer specific.

Instead, we performed two additional analyses with more in-depth characterization of the biological features identified by Griffin. First, we identified differentially expressed TFs between blood and breast cancer using gene expression data from the UCSC Xena tool; we further filtered the list to remove TFs that had extensive site overlap with TFs expressed in the opposite tissue type (Methods). For the differential TFs (35 BRCA and 22 blood), we found that many, but not all, had accessibility signals that were correlated with cfDNA tumor fraction, and that the correlations were in the expected direction suggesting the ability to predict TF activity from nucleosome profiles for a subset of TFs. We have added this to the Results:

“Next, in the cfDNA samples, we systematically analyzed differentially expressed TFs between blood cells and breast cancer (Methods, Supplementary Data 4). We found that central coverage and tumor fraction were correlated for a subset of these TFs (11 of 35 cancer and 15 of 22 blood TFs, adjusted p-value < 0.05), most correlations were in the expected direction, and that these correlations increased for blood-specific TFs after GC correction (Supplementary Fig. 4a).”

(Results, lines 157-162)

Second, we performed an analysis of covariance (ANCOVA) to identify TFBS features that were differentially accessible between ER+ and ER- samples while accounting for tumor fraction as a covariate. We found that key differential TFs such as FOXA1, GATA3, and ESR1 were among the most differential factors suggesting that Griffin is able to predict the activity of these factors from nucleosome profiles. This is described in the Results:

“First, we inspected the Griffin profiles at TFBSs for key factors, including ESR1, FOXA1, and GATA3, which are known to be associated with ER positive tumors.⁴⁹ We observed that these TFBSs were more accessible in cfDNA samples from patients with ER+ metastases compared to ER-; central coverage was significantly lower in ER+ samples after accounting for tumor fraction (ANCOVA p-value < 3.8×10^{-2} , Supplementary Fig. 9, Supplementary Data 9).”

(Results, lines 234-239)

See also Supplementary Fig. 4a, Supplementary Fig. 9, Supplementary Data 3, Supplementary data 9

3. The selection of the ER+/- specific sites in the TCGA ATAC-seq data is overly permissive and a non-parametric test does not correspond to the most appropriate statistical framework, in my opinion. An abs(log2 fold-change) of 0.5 seems very lenient to call subtype-specificity. It is not clear if the authors accounted for technical covariates were in the differential analyses (sample sequencing depth, etc). The authors should perform differential accessibility analyses using tools designed for count-based sequencing data, that account for technical factors (e.g. DESeq2). By using a more appropriate statistical framework and stringent cutoffs, the authors should be able to identify subtype-specific accessible regions with higher confidence, which will likely reflect in improved performance for Griffin to separate between ER subtypes (particularly in the low Tx samples).

Thank you for the suggestion to use DESeq2. We now use DESeq2 to identify the differential ATAC-seq sites. We also evaluated the performance for ER status classification using various cut-offs of the adjusted p-value and log2 fold change from the DESeq2 results. We found that the sites selected by the various cut-offs had minor performance differences in the classification. The final DESeq2 results selected was based on adjusted p-value cut-off of 5×10^{-4} and a log2 fold-change cutoff of 0.5, which resulted in 28,170 ER+ and 41,712 ER- subtype-specific sites. For input into the Griffin analysis, we further categorized the sites based on overlap with hematopoietic open chromatin sites, resulting in 18,240 ER+ and 19,347 ER- sites that were not shared with hematopoietic-specific sites and 9,930 ER+ and 22,365 ER- sites that were shared with hematopoietic open chromatin sites.

See Results (lines 245-254), Methods (lines 723-755), Figure 4a-b, Supplementary Fig. 11, Supplementary Fig. 12

Minor issues

4. The cfDNA dataset used by the authors for the analyses in Fig. 3a-b and Sup. Fig 3 a-d is well-balanced when considering healthy controls vs cancer patients ($n=215$ vs $n=208$). However, the dataset becomes highly imbalanced when evaluating performance in the specific subtypes (e.g. $n=12$ lung cancer patients). This can affect the ROC-AUCs interpretation. The authors should include precision-recall curves and AUCs for these comparisons, which are more robust to class imbalance. The same also applies for Fig 4c and Sup. Fig 5.

Thank you for suggesting the use of precision-recall curves, which may be better for evaluating the performance of a classifier in highly imbalanced populations. However, unlike ROC curves, precision-recall curves (PR curves) are influenced by the frequency of positive samples in a population, because precision is influenced by the total number of false positives which will increase relative to the number of true positives when the frequency of positive samples is low (Saito and Rehmsmeier, 2015, PLOS ONE; PMID: 25738806).

$$precision = \frac{true\ positives}{true\ positives + false\ positives}$$

Since the frequency of positive samples in this study represents the number of samples chosen for sequencing, not the true proportion of cancer patients in the population, this will impact the PR curves.

To illustrate this, we could imagine that Cristiano et al. had sequenced only 55 cancer samples and 215 HD for the DELFI cohort (20% cancer), in this case, the ROC curve would be mostly unchanged from the actual dataset (208 cancer samples vs. 215 HD), while the PR curve would show much lower performance. If the dataset contained only 12 cancer samples (5%), the PR curve would drop even further while the ROC curve does not change.

Similarly, the frequencies of different cancer types in the DELFI dataset influence the PR curves, but not the ROC curves, for the individual cancer types. However, the frequencies of each cancer type in the dataset are not representative of the population at large (ex. pancreatic cancer is rarer than colorectal and lung cancer but has more samples than either of those types).

Because of these differences between the study dataset and the true population, we believe that ROC curves, which are not impacted by the differences in positive class frequency, are still suitable for presenting the results in this study.

5. The authors should include diagnostic plots of the differential analyses (e.g. M-A plots) to allow better evaluation of the data normalization and differential accessibility results.

We have included M-A plots for the ATAC-seq differential analysis that was part of the DESeq2 analysis in Supplementary Fig. 11d.

6. If the authors wish to make more informative comparison with the Ulz pipeline, they may consider running it using the same fragment sizes as used by Griffin. This will allow to determine how much the GC content bias correction and the usage of smaller fragments sizes each contribute for the improved performance in Griffin.

The published version of the Ulz pipeline is hard-coded to use all fragments directly from the aligned BAM file. To demonstrate the advantages of GC content bias correction accounting for fragment sizes, we now provide a more direct comparison between Griffin and the Ulz pipeline by showing the Griffin performance when using 35-500bp fragments (which encompass the vast majority of cfDNA fragments).

For ER subtyping, we observed that using 35-500bp fragments for Griffin out-performed the Ulz pipeline (0.883 vs. 0.551 AUC). In turn, Griffin using nucleosomal-sized fragments out-performed Griffin with 35-500bp fragments (0.892 vs. 0.883 AUC)

See Results (lines 241-243), Supplementary Fig. 10

Furthermore, the inclusion of shorter fragments had benefits for deeper coverage WGS (~20x) likely due to the increased abundance for shorter fragments in the sequencing. See Results (lines 129-133), Supplementary Fig. 2

Also see Responses to Reviewer #2, Comment #9 and Reviewer #3, Comment #3.

7. For future iterations of the Griffin pipeline, I believe the sensitivity could be further improved by including features that are more specific to cancer subtypes of interest (e.g. TFBS from ChIP-seq experiments comparing ER+ to ER- tumors).

Thank you for this suggestion, we have added this to the Discussion.

**“It may be possible to improve performance of ER subtyping for lower tumor fraction samples with additional sequencing depth, using TFBSs identified directly from ER+/- tumors, or joint analysis of multiple cfDNA timepoints from the same patient.”
(Discussion, lines 347-349)**

8. It is not appropriate to call the differentially accessible regions in the ER+/- as "subtype-specific". This would imply that they are only accessible in one but not in the other subtype, which does not correspond to the differential analyses performed by the authors (particularly considering the 0.5 log2 fold-change cutoff currently used).

We have changed the text to state that these are differentially accessible sites between ER+ and ER- not necessarily specific.

9. The authors should refrain from using subjective descriptors to their results, such as "excellent" (line 7) or "great" (line 170).

These adjectives have been removed.

10. I believe a reference to panel 4e is missing from line 244 in the main text.

Figure 4 panels have now changed, and the text reflects this change.

REVIEWER COMMENTS

Reviewer #1 (Remarks to the Author):

This is a revised manuscript describing a new framework for inference of cancer subtype from plasma DNA analysis using 0.1x WGS. In particular, the authors demonstrate the ability to predict ER status in patients with metastatic breast cancer with high accuracy.

While the revisions better clarify performance of the Griffin framework, the abstract overstates the findings of this manuscript and its expected clinical/real-world performance and needs to be revised significantly.

It remains unclear what the true clinical relevance of this advance may be as demonstrated here. Accurate prediction of ER status using Griffin requires $\geq 5\%$ tumor fraction. Authors contend that $\sim 30\%$ of patients who present with metastatic breast cancer have $> 10\%$ tumor fraction but this does not account for the observation that tumor fractions in plasma DNA are generally lower for ER+ breast cancer patients than patients with TNBC. In addition, prediction of ER status seems to be a particularly accessible example where enough sites across the genome can be integrated. The real-world expected performance of ER status prediction in patients with MBC will likely be much lower and there is no clear demonstration that these findings will be generalizable into other cancer types and subtypes. The abstract omits the very significant detail that patients and samples were pre-selected for this analysis to have at 5% tumor fraction, which in this reviewer's observation is a much higher ctDNA level than observed routinely at presentation in ER+ MBC patients.

A key claim of novelty is GC correction. While this report is a useful evaluation of the impact of GC bias on fragmentation metrics, the actual impact of this on ROC performance of Griffin for subtype classification using ULP-WGS is very minor.

Cancer detection performance reported in the abstract is an AUC of 0.94 for early stage cancers, and the use of 0.1x WGS is implied. However, ULP-WGS performance in text is an AUC of 0.69 for all stages (in the LUCAS cohort). A more accurate representation of the cancer detection performance using ULP-WGS data across these cohorts is needed in the abstract.

Reviewer #2 (Remarks to the Author):

The authors have addressed most of my concerns in their response and revised manuscript. However, I have one remaining and key concern relating to the robustness (accuracy on unseen samples/cohorts) of their approach/model/features. In the revised manuscript, the authors convincingly demonstrate that their ER-subtype model can generalize across distinct breast cancer cohorts. However, they also note that a large cohort of lung cancer samples (Cristiano et al. Nature, 2019; PMID: 31142840) demonstrate a "batch-effect" that prevents the evaluation of their lung cancer detection model on these samples. When looking closer at this putative batch-effect (Suppl. Fig. 8), the authors note that the effect is primarily coupled to their "central-coverage" features.

This is a problem for people wanting to use Griffin's existing models or exploring to build their own models using the Griffin framework. How would you determine if your new/unseen samples (or training samples) suffer from the same batch effect? The authors should spend more time investigating the potential underlying source of this batch effect and ways it could be mitigated. Is the effect caused by their GC correction approach? Is the effect coupled to sequencing parameters? Is the effect coupled to specific TF's? Can it be observed in other cohorts? Can it be mitigated by "batch-correction" approaches like ComBat?

Reviewer #3 (Remarks to the Author):

The authors are to be commended on the additional data and revisions that have significantly strengthened the manuscript. The majority of my previous concerns have now been satisfactorily addressed, with two minor comments:

1. The authors now provide a systematic performance comparison between inclusion and exclusion of copy number correction. For cancer detection, they observed similar results between including and excluding CNA correction. For cancer detection, this was only assessed on 1-2x WGS coverage data (Supp Figure 7). Given that a large focus of the manuscript is on the use of Griffin for cancer detection with ultra low pass WGS it would be beneficial to see the impact of CNA correction on 0.1 x WGS coverage data.

2. The authors state that the DELFI and LUCAS cohorts had notable batch effects which prevented use of the same model on both cohorts. In the description of the training and testing of the new model in the LUCAS cohort, the methods state that "To get this final model, we performed PCA on the full LUCAS cohort and extracted 35 features that explained 80% of the variance" line 1283. It would be important to clarify that the full LUCAS cohort referred to here, does not include the samples within the LUCAS validation cohort.

Reviewer #4 (Remarks to the Author):

The authors performed a thorough job of addressing mine and the other reviewers' comments. I think the manuscript is now much improved and I only have a few very minor comments.

It is interesting to see that the accuracy for the low TFX predictions is highest on the more stringent q-val cutoff in Sup. Fig. 11d. This is consistent with what I would expect from the MA plot in Supp. Fig. 11c, which suggests that even with DESeq2 the data is not as cleanly modeled. This is likely because it's aggregating from multiple studies/centers. It is not clear if the authors used any co-variables in their DESeq2 models, such as batch (e.g. study of origin) or any other potentially relevant patient data (e.g. age). This could potentially improve the underlying DESeq2 model and highlight more diagnostic features between the subtypes to improve predictions. Please note that I am not requesting this analysis - I think the authors did adequate exploration with the differential analyses parameters.

I thank the authors for their informative discussion regarding ROC curve usage when only a subset of the entire population is used for testing, which I will keep in mind for the future.

Minor points

Some of the supplementary figures seem to have very low resolution - Supp. Fig. 5, in particular, is barely clear enough to read the text. This might be an issue with the conversions during upload, so the authors should check the final uploaded files to make sure they all look sharp.

Supp Fig 6 should include explanation to the terms TFX, CA, and HD in the legend to facilitate interpretation by the reader. TFX is explained in one of the main figures, but I couldn't find the others.

We thank the reviewers again for their careful reading of the revised manuscript. These additional comments provide excellent suggestions. We have addressed them all below and believe that the manuscript has benefitted from these suggestions. Please see the point-by-point responses below.

Reviewer #1 (Remarks to the Author):

This is a revised manuscript describing a new framework for inference of cancer subtype from plasma DNA analysis using 0.1x WGS. In particular, the authors demonstrate the ability to predict ER status in patients with metastatic breast cancer with high accuracy.

While the revisions better clarify performance of the Griffin framework, the abstract overstates the findings of this manuscript and its expected clinical/real-world performance and needs to be revised significantly.

It remains unclear what the true clinical relevance of this advance may be as demonstrated here. Accurate prediction of ER status using Griffin requires $\geq 5\%$ tumor fraction. Authors contend that $\sim 30\%$ of patients who present with metastatic breast cancer have $>10\%$ tumor fraction but this does not account for the observation that tumor fractions in plasma DNA are generally lower for ER+ breast cancer patients than patients with TNBC. In addition, prediction of ER status seems to be a particularly accessible example where enough sites across the genome can be integrated. The real-world expected performance of ER status prediction in patients with MBC will likely be much lower and there is no clear demonstration that these findings will be generalizable into other cancer types and subtypes. The abstract omits the very significant detail that patients and samples were pre-selected for this analysis to have at 5% tumor fraction, which in this reviewer's observation is a much higher ctDNA level than observed routinely at presentation in ER+ MBC patients.

Thank for raising these considerations and interpretation of the approach. We have clarified in the abstract that the performance applies when $> 5\%$ tumor fraction for metastatic breast cancer.

“Cell-free DNA (cfDNA) has the potential to inform tumor subtype classification and help guide clinical precision oncology. Here we developed Griffin, a new framework for profiling nucleosome protection and accessibility from cfDNA to study the phenotype of tumors using as low as 0.1x coverage whole genome sequencing (WGS) data. Griffin employs a novel application of a GC correction procedure tailored to variable cfDNA fragment sizes, which generates a better representation of chromatin accessibility and improves the accuracy of cancer detection and tumor subtype classification. We applied Griffin for the first demonstration of estrogen receptor (ER) subtyping in metastatic breast cancer from cfDNA. We predicted ER subtype (AUC=0.89) in 139 patients with at least 5% detectable ctDNA, and validated performance in independent cohorts (AUC=0.96). In summary, Griffin is a framework for accurate tumor subtyping and can be generalizable to other cancer types for precision oncology applications.”
(Abstract)

A key claim of novelty is GC correction. While this report is a useful evaluation of the impact of GC bias on fragmentation metrics, the actual impact of this on ROC performance of Griffin for subtype classification using ULP-WGS is very minor.

The reviewer is correct that impact of GC correction on performance of subtype classification (ULP-WGS) is minor (0.87 AUC uncorrected vs 0.89 corrected). We have added this limitation to the Discussion to clarify where benefits were gained by GC-correction.

**“We observed improved performance after GC-correction consistently for all analyses, suggesting the benefit of the approach, although this improvement was minor for ER status prediction in ULP-WGS data.”
(Discussion – lines 335-336)**

Cancer detection performance reported in the abstract is an AUC of 0.94 for early stage cancers, and the use of 0.1x WGS is implied. However, ULP-WGS performance in text is an AUC of 0.69 for all stages (in the LUCAS cohort). A more accurate representation of the cancer detection performance using ULP-WGS data across these cohorts is needed in the abstract.

The reviewer brings up an important concern. After careful consideration, due to the word count limitations and to avoid confusion and misattribution of performance for various cohorts and sequencing coverages, we have elected to exclude explicit reference to AUC values in the abstract. This will better emphasize the unique results of the breast cancer ER status prediction.

We thank the reviewer for providing these important suggestions, which has helped to provide a more focused and better presentation of the results in this study.

Reviewer #2 (Remarks to the Author)

The authors have addressed most of my concerns in their response and revised manuscript. However, I have one remaining and key concern relating to the robustness (accuracy on unseen samples/cohorts) of their approach/model/features. In the revised manuscript, the authors convincingly demonstrate that their ER-subtype model can generalize across distinct breast cancer cohorts. However, they also note that a large cohort of lung cancer samples (Cristiano et al. Nature, 2019; PMID: 31142840) demonstrate a “batch-effect” that prevents the evaluation of their lung cancer detection model on these samples. When looking closer at this putative batch-effect (Suppl. Fig. 8), the authors note that the effect is primarily coupled to their “central-coverage” features.

This is a problem for people wanting to use Griffin’s existing models or exploring to build their own models using the Griffin framework.

Thank you for raising these potential concerns and we appreciate your suggested inquiries into addressing them. We provide responses and/or additional analysis for each of the questions raised.

First, we would like to clarify that the batch effect was observed between the DELFI and LUCAS/LUCAS validation cohorts and not within the DELFI cohort itself. We have clarified in the text – note that this clarification also incorporates some of the technical clarifications we concluded after addressing questions below.

**“There was a notable batch effect between the DELFI and LUCAS cohorts in the initial fragment size distributions and Griffin coverage profiles before and after GC correction, which prevented use of the same model on both cohorts (Methods, Supplementary Fig. 8, Supplementary Data 7)”
(Results - lines 214-217)**

Supplementary Fig. 8

Supplementary Fig. 8: Principal component analysis (PCA) on Griffin features for cancer detection cohorts before GC correction (**a**) and after GC correction (**b**). For each cancer detection cohort (DELFI, LUCAS, and LUCAS validation) Griffin analysis was performed on 30,000 TFBS each for 270 TFs and 3 features (central coverage, mean coverage, and amplitude) were extracted from each profile for a total of 810 features. Top row, a PCA was performed on all features for all samples and the top two components were plotted for healthy samples from all three cohorts (left), cancer samples (middle) and all samples (right). The DELFI cohort clustered away from the other cohorts indicating systematic difference between the DELFI cohort and other cohorts. Next, PCA was performed separately on each of the 3 feature types: central coverage (second row), mean coverage (third row), and amplitude (bottom row) which revealed that the difference between the DELFI cohort and other cohorts was primarily due to differences in the central coverage. Percentage of variance explained by each PC is labeled on the axes. **(c)** Mean normalized fragment size profiles for the three cohorts. Shading indicates IQR. **(d)** PCA of the fragment size profiles in the three cohorts. Top two PCs are shown for healthy samples (left), cancer samples (middle), and all samples (right).

How would you determine if your new/unseen samples (or training samples) suffer from the same batch effect?

While we understand the reviewer's concern that an ideal model should work for all datasets, unfortunately, it is challenging in the datasets that we are using for cancer detection from cfDNA. Although our trained cancer detection models are not compatible with all datasets, we still present a full workflow/framework so that users can extract features from cfDNA and train their own model, which we believe is an important contribution and deliverable. When analyzing new samples, users should first apply the Griffin pipeline to extract features. Users can then test the performance of our existing machine learning models, and if batch effects contribute significantly to reduced performance, they can then train a new model on their specific cohort/platform using our framework. This is a common approach and limitation that many other applications employing machine learning also face. Furthermore, this will allow flexibility for users to test various different machine learning algorithms and feature sets that suit their studies and experiences. Future prospective collection of samples using common sample workflows will provide a better assessment of the true performance for cancer detection. We have highlighted this limitation in the main text.

“While the GC correction strategy was able to reduce inter-sample variability, we found that it was not able to eliminate batch effects between datasets potentially caused by different cfDNA processing and sequencing workflows, thus preventing cancer detection models from being compatible across all datasets. However, Griffin provides a framework to extract cfDNA features, enabling users to train models on new datasets, as we showed with the LUCAS and validation cohorts. Griffin can be applied to future large prospective studies using standardized plasma collection and workflows to carefully assess the performance of cancer detection in real clinical scenarios.”

(Discussion - lines 336-343)

The authors should spend more time investigating the potential underlying source of this batch effect and ways it could be mitigated.

Thank you for this suggestion. We examined a number of biological covariates including cancer status, sex, cancer type, tumor fraction, and age. None appeared to explain the batch effect observed in central coverage features between the DELFI and LUCAS/LUCAS validation cohorts. Please see figure attached below.

Response to Reviewer comments for *Doebley et al. (NCOMMS-21-33156A)*

We also examined the fragment size profiles (using picard CollectInsertSizeMetrics) in the DELFI, LUCAS, and LUCAS validation cohorts. We observed that even in healthy controls, the LUCAS cohort had, on average, more nucleosome sized fragments and fewer short fragments compared to the DELFI cohort. We applied PCA to the fragment size distribution and observed that the DELFI samples clustered slightly away from the other samples. This result indicated that there were inherent differences in the fragmentation between the cohorts in the sequencing data. We have included this comparison in Supplementary Fig. 8 and noted it in the Results section (Results – line 214-217).

Is the effect caused by their GC correction approach?

Based on our observations above that the fragment size distributions were already different from the sequencing data prior to normalization by Griffin, we concluded that the batch effects were not introduced by Griffin itself. To further show this, we performed a PCA of the features without GC correction and observed that the same batch effect is present in the data. We have included this analysis to Supplementary Fig. 8 and noted it in the Results section (Line 214-217).

Is the effect coupled to sequencing parameters?

While we did not generate the sequencing data used for the cancer detection analysis, we can only *speculate* about what may have caused the batch effect by reading the Methods for these studies. There do not appear to be any differences in the sequencing parameters between the cohorts, although it is possible that there were small differences not fully described. Prior to sequencing, the number of PCR amplification cycles may have been different. Four cycles were used for the LUCAS dataset after fine tuning to reduce GC biases, but the number of cycles was not listed for the DELFI dataset. Additionally, the sample collection protocols may have been different between the cohorts. For example, there were different lengths of time the samples were stored in EDTA tubes before processing ('immediate processing' in DELFI, vs 'within two hours' for LUCAS, incubation in EDTA is known to effect fragmentation profiles). Also, different speeds for the centrifugation step were used between the cohorts (an initial spin of 800g plus a second spin of the plasma portion at 18,000g for DELFI vs a single spin at 2330g for LUCAS and an initial spin at 1500-3000g plus a second spin of the plasma portion for the LUCAS validation).

Is the effect couple to specific TF's?

We inspected the Griffin features between DELFI and LUCAS cohorts for 270 TFs using a Kolmogorov-Smirnov (KS) test with Benjamini-Hochberg FDR correction to determine whether TFs had a significantly different distribution. We found that 255/270 TFs had significantly different distributions in the central coverage value between healthy donors in the two cohorts, suggesting the batch effect is widespread across most TFs and consistent with the global differences in fragment sizes between the cohorts. For comparison, the amplitude feature did not show a clear batch effect by PCA, with only 17 significantly different TFs. We have added data to new Supplementary Data 7.

Can it be observed in other cohorts?

We did not observe a batch effect across the 4 breast cancer cohorts for the features used for ER status prediction. We have now added an additional panel to Supplementary Fig. 13 (new panel a) illustrating that batch effect is not likely the key contributor to signal differences. Rather, we observed biological features of ER status and tumor fraction, as

shown with the top two PCs for these cohorts. We have modified the text to note this observation.

“Using PCA, we did not observe batch effects between the cohorts, but rather signals could be attributed to the known ER status (by metastatic tumor IHC) and estimated tumor fraction (Supplementary Fig. 13a).”
 (Results - lines 267-269)

Supplementary Fig. 13: MBC validation set performance (a) Principal component analysis (PCA) on Griffin features for breast cancer samples from the initial ULP-WGS cohort and three validation cohorts. For each cohort Griffin analysis was performed on differential ATAC seq sites identified by DESeq2 using the 5×10^{-4} adjusted p-value cutoff. 3 features were extracted from each profile for a total of 12 features. A PCA was performed on all 12 features (first column), central coverage features only (second column), mean coverage features (third column), or amplitude features (fourth column). This PCA was then colored by cohort (top row) to look for batch effects, but batch effects were not observed in the top two principal components. The PCA was also colored by ER status (second row), demonstrating that the first PC (PC_0, x axis) appears to correspond to status. Finally, the PCA was colored by tumor fraction (third row) indicating that the second PC (PC_1, y axis) corresponds to tumor fraction. Percentage of variance explained by each PC is labeled on the axes.

Can it be mitigated by “batch-correction” approaches like ComBat?

Thank you for this suggestion. We attempted to apply batch correction to the LUCAS and LUCAS validation cohorts using comBat with non-parametric adjustment and using the DELFI cohort as the reference batch:

```
ComBat (dat=data, batch=batch, mod=NULL, par.prior=FALSE,
ref.batch='DELFI')
```

We then applied the model trained on the DELFI cohort to the batch corrected LUCAS and LUCAS Validation cohorts. Although the comBat correction appears to have removed the batch effect observed by PCA (shown below for ‘all features’ and ‘central coverage features’), it did not improve the performance of the model over the non-batch corrected version. Please see attached figure below.

This result may suggest that features that were important for cancer detection in the DELFI cohort may not necessarily be the same in the LUCAS cohorts for reasons we were unable to determine from our investigation.

PCA after comBat batch correction:

AUC (95% CI)	Trained on LUCAS cohort (bootstrapped cross validation used for testing on LUCAS cohort)		Trained on DELFI cohort No batch correction		Trained on DELFI cohort Batch corrected LUCAS and LUCAS validation cohorts	
	LUCAS 1-2x WGS	Validation 1-2x WGS	LUCAS 1-2x WGS	Validation 1-2x WGS	LUCAS 1-2x WGS	Validation 1-2x WGS
I	0.57 (0.31-0.79)	0.83 (0.73-0.91)	0.57 (0.43-0.72)	0.72 (0.58-0.84)	0.55 (0.40-0.69)	0.70 (0.56-0.82)
II	0.77 (0.40-0.98)	0.86 (0.70-0.97)	0.68 (0.40-0.91)	0.76 (0.50-0.96)	0.69 (0.43-0.90)	0.74 (0.47-0.96)
III	0.79 (0.66-0.90)		0.73 (0.62-0.83)		0.72 (0.61-0.82)	
IV	0.79 (0.68-0.87)		0.74 (0.66-0.82)		0.71 (0.63-0.79)	
III-IV		1.00 (1.00-1.00)		0.95 (0.84-1.00)		0.96 (0.86-1.00)
overall	0.76 (0.67-0.83)	0.86 (0.78-0.91)	0.71 (0.65-0.78)	0.76 (0.65-0.85)	0.69 (0.63-0.76)	0.74 (0.63-0.84)

Reviewer #3 (Remarks to the Author)

The authors are to be commended on the additional data and revisions that have significantly strengthened the manuscript. The majority of my previous concerns have now been satisfactorily addressed, with two minor comments:

We thank the reviewer for their very helpful suggestions for the original manuscript that help us to significantly strengthen the manuscript.

1. The authors now provide a systematic performance comparison between inclusion and exclusion of copy number correction. For cancer detection, they observed similar results between including and excluding CNA correction. For cancer detection, this was only assessed on 1-2x WGS coverage data (Supp Figure 7). Given that a large focus of the manuscript is on the use of Griffin for cancer detection with ultra low pass WGS it would be beneficial to see the impact of CNA correction on 0.1 x WGS coverage data.

Thank you for this suggestion. We have now added a panel to Supplementary Fig. 7a showing the impact of CNA correction on cancer detection in ULP data. We found that explicit CNA correction did not improve cancer detection performance for 0.1x ULP-WGS data. We have attached the pertinent panel below. Supplementary Fig. 7 now also contains the consolidated performance metrics for the other configurations and comparisons of Griffin on ULP data into this panel.

Supplementary Fig. 7: Evaluation of various configurations and comparisons of Griffin for cancer detection. (a) Boxplots of the AUC values for 1,000 bootstrap iterations of the logistic regression classifier using various configurations and comparisons of Griffin on the DELFI cohort (1-2x WGS data) and downsampled DELFI cohort (0.1x WGS) grouped by stage. The boxed range represents the median ± IQR, whiskers represent the range of the non-outlier data (maximum extent is 1.5x the IQR). Outliers are shown as grey diamonds. (vii-xi) Performance of selected configurations on 0.1x WGS data.

2. The authors state that the DELFI and LUCAS cohorts had notable batch effects which prevented use of the same model on both cohorts. In the description of the training and testing of the new model in the LUCAS cohort, the methods state that “To get this final model, we performed PCA on the full LUCAS cohort and extracted 35 features that explained 80% of the variance” line 1283. It would be important to clarify that the full LUCAS cohort referred to here, does not include the samples within the LUCAS validation cohort.

This is correct, in order to get the features that explained 80% of the variance, the PCA was performed on only the LUCAS cohort, not including the LUCAS validation cohort. The same PCA transformation was then applied to the validation cohort. We have added text to the methods to clarify this:

**“To get this final model, first, we performed PCA on the full LUCAS cohort (not including the LUCAS validation cohort) and extracted 35 features that explained 80% of the variance in that cohort.”
(Methods – lines 873-875)**

Reviewer #4 (Remarks to the Author)

The authors performed a thorough job of addressing mine and the other reviewers' comments. I think the manuscript is now much improved and I only have a few very minor comments.

We thank the reviewer for their thoughtful comments that helped to improve the manuscript.

It is interesting to see that the accuracy for the low TFX predictions is highest on the more stringent q-val cutoff in Sup. Fig. 11d. This is consistent with what I would expect from the MA plot in Supp. Fig. 11c, which suggests that even with DESeq2 the data is not as cleanly modeled. This is likely because it's aggregating from multiple studies/centers. It is not clear if the authors used any co-variates in their DESeq2 models, such as batch (e.g. study of origin) or any other potentially relevant patient data (e.g. age). This could potentially improve the underlying DESeq2 model and highlight more diagnostic features between the subtypes to improve predictions. Please note that I am not requesting this analysis - I think the authors did adequate exploration with the differential analyses parameters.

We didn't use any covariates in the DESeq2 analysis and have clarified this in the methods. However, we thank you for the suggestion and we will definitely include relevant covariates when running DESeq2 in the future.

**“The software was run using default settings described in the ‘quick start’ guide with no co-variates.”
(Methods – line 746-747)**

I thank the authors for their informative discussion regarding ROC curve usage when only a subset of the entire population is used for testing, which I will keep in mind for the future.

Thank you for raising the initial comment and for this note. The co-authors and the lab learned a lot on the discussion of this topic.

Minor points

Some of the supplementary figures seem to have very low resolution - Supp. Fig. 5, in particular, is barely clear enough to read the text. This might be an issue with the conversions during upload, so the authors should check the final uploaded files to make sure they all look sharp.

Thank you for bringing this to our attention. We have now replaced Supplementary Figures with higher resolution versions.

Supp Fig 6 should include explanation to the terms TFX, CA, and HD in the legend to facilitate interpretation by the reader. TFX is explained in one of the main figures, but I couldn't find the others.

We have added these definitions to the legend for Supplemental Fig. 6.

REVIEWERS' COMMENTS

Reviewer #2 (Remarks to the Author):

The authors have addressed all my remaining concerns, great work!

Reviewer #3 (Remarks to the Author):

The authors are to be commended on their revisions. They have satisfactorily addressed all of my previous concerns and suggestions and these changes have significantly strengthened the manuscript.